# Density dependence of songbird demographics in grazed sagebrush steppe

**Kayla A. Ruth** [1,2]*, **Lorelle I. Berkeley**[3,4], **Kaitlyn M. Strickfaden**[1], **Victoria J. Dreitz**[1]

**1** Wildlife Biology Program and Avian Science Center, W.A. Franke College of Forestry and Conservation, University of Montana, Missoula, Montana, United States of America, **2** Department of Animal and Rangeland Sciences, Oregon State University, Corvallis, Oregon, United States of America, **3** Montana Fish, Wildlife & Parks, Helena, Montana, United States of America, **4** SWCA Environmental Consultants, Salt Lake City, Utah, United Stated of America

☯ These authors contributed equally to this work.
* kayla.ruth@oregonstate.edu

**Data Availability Statement:** All relevant data are within the paper and its Supporting Information files.

**Funding:** LIB - Federal Aid in Wildlife Restoration Grant F16AF00294 to Montana Fish Wildlife and

## Abstract

Sagebrush steppe is one of the most threatened ecosystems in North America. Adult density of songbirds within sagebrush steppe is a metric used to evaluate conservation actions. However, relying on only adult density to guide conservation may be misleading. Information on how conservation actions influence the nest density and nest survival of songbird species, in addition to adult density, are needed. We evaluated the relationships between nest density, nest survival, and adult density of Brewer's sparrow (*Spizella breweri*) and vesper sparrow (*Pooecetes gramineus*) over 3 breeding seasons in central Montana. Our findings suggest that adult pairs of both species were often present in higher numbers than nests, and this relationship was most prominent for Brewer's sparrows. However, our results do not support density dependence when considering nest survival. This discrepancy suggests that songbirds may not breed every year and that density dependence may be operating on nest densities within these populations differently than we examined. This study provides information on relationships between population demographics for 2 songbird species in grazed sagebrush steppe that will improve monitoring and management activities of conservation efforts.

## Introduction

Successful wildlife management and conservation require a foundational understanding of mechanisms influencing wildlife populations, including population regulation [1]. Ecologists have attempted to understand the mechanisms limiting populations for decades [2]. The most commonly observed regulatory process controlling population dynamics is density dependence [1, 3–6]. Density dependence occurs when the density of individuals affects the growth rate of the population through birth, death, and movement rates [2, 7]. For instance, density dependence affects survival and breeding productivity in barn swallows (*Hirundo rustica*: [8]). Previous studies have identified two potential primary mechanisms for density-dependent population regulation [9, 10]. The first hypothesized mechanism is that density dependence is

Parks. VJD - Montana Fish Wildlife and Parks awarded grant W-158-R-1 to University of Montana fwp.mt.gov www.fws.gov/program/wildlife-restoration The funders had no role in study design, data collection and analysis, decision to publish, or preparation of the manuscript.

**Competing interests:** The authors have declared that no competing interests exist.

driven by habitat and breeding site selection, and available habitat is heterogeneous in terms of quality [9, 11]. Higher quality habitat sites are associated with higher vital rates, including survival and reproduction [11]. However, as population size surpasses what optimal sites can support, suboptimal sites become occupied and result in reduced vital rates; therefore, growth rates decreases [12]. The second hypothesized mechanism focuses on individual behavior [9]. Increased intraspecific aggression, competition, and interference due to high population densities can decrease vital rates within a population, resulting in decreased population growth rates [1, 3, 4, 9, 10, 13]. Investigating the extent of density dependence effects on wildlife populations provides a more comprehensive understanding of population regulation that can be used to inform conservation actions.

Sagebrush steppe songbirds are one of the fastest declining guilds in North America [14, 15]. Three interrelated demographic parameters affecting the number of individuals in avian populations are adult density, nest density, and nest survival. Studies on birds have investigated possible density-dependent relationships between adult density and nest survival [e.g., 16, 17, 18]. Nest survival is suggested to be density-dependent, where lower nest survival is a trade-off between improved resources and increased competition [19, 20]. Most studies assume that adult density serves as a proxy for nest density [e.g., 21, 22], especially since adult density is often used to evaluate the effects of management actions on species. However, few studies have examined whether adult density accurately reflects nest density [23–25]. Competition for nest sites may prevent some adults from breeding [19, 20]. Thus, management at a site might have detrimental effects on a population, but the most common metric for assessing this, counts of adults, would not indicate an issue. To our knowledge, no studies have considered nest density in relation to density-dependent regulation in sagebrush songbirds. Investigating the relationships between nest density, adult density, and nest survival provides initial steps in discerning mechanisms regulating population growth that these parameters measured individually may not indicate [23].

Multiple anthropogenic threats to sagebrush steppe songbirds have been suggested, including conversion of sagebrush to agriculture [26]; fragmentation resulting from energy [27] and subdivision development [28]; conifer expansion (e.g., in Oregon and western Montana); [26, 29] and modifications such as prescribed fire, herbicides, and grazing practices that have led to exotic annual grass establishment [30]. Numerous management actions have been established to prevent further declines in sagebrush steppe songbirds through sagebrush conservation programs. One of the most notable conservation programs is the Sage Grouse Initiative, established in 2010 by the US Department of Agriculture - Natural Resources Conservation Service (NRCS). The Initiative uses a partnership-based approach that integrates socioeconomic and ecological values to improve sage-grouse habitat. It integrates private landowners into land management activities that support greater sage-grouse (*Centrocercus urophasianus*) conservation while improving rangeland sustainability for all stakeholders with objectives dependent on regional challenges [31, 32]. The land management activities to improve habitat for sage-grouse may also alter resources and habitat quality that influences population regulation of other sagebrush-obligate species. For instance, one regional objective implements domestic livestock grazing management to minimize adverse grazing effects on vegetation for sage-grouse while keeping working ranches intact to avoid further fragmentation of the sagebrush steppe from development. Consequently, the management actions of the conservation program may influence density dependent responses in sagebrush steppe songbirds by altering breeding demographic rates because of changes to the habitat.

Here, we use Brewer's sparrow (*Spizella breweri*), an obligate sagebrush species, and vesper sparrow (*Pooectes gramineus*), a generalist species [24], to assess density dependence within the context of the Sage Grouse Initiative (SGI) conservation program. We identified the effects

of nest density on nest survival across three breeding seasons and examined the relationship between nest density and adult density per season. We tested the hypothesis that nest survival is density-dependent, leading to a negative effect of nest density on nest survival, and the alternative hypothesis of no relationship between nest density and survival. We also hypothesized that adult density drives nest density, resulting in a linear association between adult pair density and nest density (i.e., 1 nest per 2 adults) per breeding season. Limited information exists on how management actions of conservation programs influence processes that regulate songbird populations. Thus, as an initial step, we explored the influence of the SGI conservation program on nest density, nest survival, and adult density, recognizing that differences in vegetation characteristics from conservation programs could affect nest sites and overall habitat selection during a breeding season. Understanding the role of conservation programs on demographic rates and whether density dependence influences any of these demographic rates is important for informing future relevant management decisions.

## Methods

### Study area

The study area encompassed 89,000 ha of sagebrush steppe Golden Valley and Musselshell counties near Roundup, Montana, USA. We sampled sites on private and public lands grazed by domestic livestock, primarily cattle (*Bos taurus*). Dominant shrubs were Wyoming big sagebrush (*Artemisia tridentata* spp. *wyomingensis*) and silver sagebrush (*Artemisia cana*), and dominant grasses were needle-and-thread grass (*Hesperostipa comata*) and western wheatgrass (*Pascopyrum smithii*). Our study area was on the edge of sagebrush steppe and grassland ecosystems in the western US. Thus, it contained less shrub and forb cover and shorter shrubs than other sagebrush steppe areas [32]. The average annual precipitation for the area is ∼36 cm [32, 33].

We collected data from 1 May– 15 July 2016–2018 on 80 randomly selected sampling plots following Institutional Animal Care and Use permit 008-18VDWB-02121. Each sampling plot was 500 x 500 m (25 ha) following other avian studies [34–37]. Our random selection process required individual plots to be > 500 m from each other, resulting in a range of 713 m to 12,511 m between plots. Of the 80 plots, 40 were privately managed pastures participating in the SGI partnership-based program in which private landowners worked collaboratively with NRCS range specialists in developing domestic livestock grazing management practices for the specific pasture (hereafter, referenced as SGI plots). The remaining 40 plots were located on private and publicly managed pastures not participating in the partnership-based program (hereafter, non-SGI) with grazing management practices developed by the private landowner or required by the public land management agency (e.g., US Department of Interior–Bureau of Land Management). We collected adult and nest data to estimate nest density, nest survival, and adult density on each plot. We examined yearly differences, land enrollment in conservation program (e.g., SGI or non-SGI), and other biotic and abiotic factors on these 3 demographic parameters (S1 Table). The biotic factors were vegetation metrics that included the proportion of ground cover of shrubs (SHR), vegetative productivity using gross primary production (GPP Mean), and vegetative structure using leaf area index (Mean LAI). Abiotic factors were mean precipitation and max and min temperature. We compared these factors between land enrollment (e.g., SGI or non-SGI). However, there were no significant differences in the covariates between land enrollment (S2 Table).

### Nest density and nest survival surveys

We surveyed plots based on distance sampling protocols [38, 39] to estimate nest density. We surveyed > 4 transects 500 m long for each plot at ∼2.5-week intervals (3–4 times over the

season; each plot 500 x 500 m [25 ha]). The distance from the transect to a nest allowed for an estimate of detection probability [38, 39]. We placed north-south transects parallel to each other every 100 m across the plot starting > 25 m from the plot edge with 2 observers walking approximately 10 m apart (each observer 5 m from the transect line). In shrub-dominated plots or areas within plots, the observers tapped the tops of shrubs with wooden dowels to flush nesting adults [40]. In plots, or areas within a plot, dominated by grasses, 2 observers dragged a 10-m-long chain between them that brushed the ground vegetation to flush nesting adults [41–43]. Vesper sparrows will nest on the ground, and we assumed the chain method would increase our ability to detect their nests. We collected geographic coordinates for each nest (a structure with ≥ 1 egg) to compute the distances between the transect line and nests. GPS units used for this project have been reported to have planimetric accuracy of 3.12 m [44]. The mean distance from transect lines to nests was 5.42 ± 3.69 m.

We initiated our nest surveys each year based on anecdotal information on the presence and behavior of species in our study area (e.g., eBird reports) and completed the surveys based on the level of breeding activity and behavior. This resulted in nest surveys being conducted from 8 May to 1 July each year. At nest discovery and subsequent visits, we recorded the species, nesting stage, and the number of eggs or nestlings. We monitored nests every ∼ 3 days, weather permitting, until evidence of success or failure. We characterized a nest as successful if ≥ 1 nestling fledged (left the nest; [45–47]). A failed nest had evidence of depredation (e.g., destroyed nest) or, in rare cases, abandonment (e.g., unhatched eggs after expected hatch date). We also located nests opportunistically (e.g., during adult surveys) and monitored their status. Opportunistic nests were included in the nest survival analysis but not in the nest density analysis because opportunistic nests did not adhere to distance sampling protocols which provide an estimate of nest detection. As birds were not individually marked, we could not verify renesting attempts or rates of multibrooding and acknowledge that this may exist within our dataset.

## Adult density surveys

We conducted count-based surveys 3–4 times throughout the season to estimate adult density on the same 80 plots we conducted nest searches. Most often, adult density surveys for a plot were conducted the day before nest surveys. On the few occasions (< 5% of the surveys), due to inclement weather, the adult density and nest surveys were not completed on consecutive days, instead they were completed 2–3 days apart.

We counted adults on plots following a dependent double-observer (DDO) survey method developed by Nichols et al. [48] except we only used visual detection as described by Golding and Dreitz [36, 37]. This method was ideal for our study area and has been used successfully in open, arid environments [35, 36, 49]. We conducted surveys along a U-shaped transect within a plot that allowed two observers to survey ≤125 m on each side of the transect (S1 Fig; also see [34, 35]). This distance was chosen as ≥95% of visual songbird detections occur within 125 m of the transect line [36, 48]. The surveys required two observers, a primary and a secondary, with different roles [48]. The primary observer walked roughly 5 m ahead of the secondary observer communicating visual observations of adults. The secondary observer recorded the primary observer's observations and observations that the primary observer missed that they observed. Observers used auditory cues and binoculars to assist with detection. To ensure observations took place within plot boundaries and ensure that individuals were not double-counted, audio detections required visual observations as well. Surveys took place between sunrise (approximately 0530 Mountain Standard Time [MST]) and 1100 MST and were not conducted during heavy precipitation or wind speeds > 15 mph.

## Nest density analysis

We calculated nest density per plot and year based on line transect distance sampling [38, 50] in program R (ver. 3.5.1, www.r-project.org, accessed 15 Aug 2016) with package 'Distance'. Distance sampling along a line transect generates a detection function that requires $\geq 2$ data points to determine how the probability of detection varies with distance from the transect line [38, 39]. Thus, the dataset included plots with $\geq 2$ nests detected per plot for each species annually. We developed a suite of candidate models using half-normal, uniform, and hazard-rate key functions and cosine, simple polynomial, and Hermite polynomial adjustment terms [50]. We included abiotic (i.e., mean precipitation and average maximum temperature) and biotic factors (shrub cover, mean LAI, and mean GPP) on detection in our models for each year and species (S3 Table). We analyzed nest density in groups based on year and land enrollment in conservation program (SGI or non-SGI; Table 1 and Fig 1). Due to small sample size, we grouped estimates by year and land management enrollment. Most nests (95%, 118 out of 124 nests) were located using the dowel search method; thus, search method was not included as a detection covariate. Akaike Information Criterion for small sample sizes (AIC$_c$; [51]) was used to select the top model. The top model for predicting nest density for each year and species was used to estimate nest density at the plot level and as species-specific nest density metric for nest survival analyses.

## Nest survival analysis

We used a logistic exposure model [52, 53] to estimate daily nest survival. Daily survival rates (DSR) can vary between different periods within the nesting stage (e.g., eggs or young); thus, we included a variable for stage as either eggs or nestlings [45, 54, 55]. We included year to account for temporal variation. Additionally, a random effect was included to account for any unexplained variation in the plots [52, 56].

To investigate the effects of nest density on nest survival, we included a estimate of species-specific nest density as a covariate, which reflected the estimated density at the plot-level. Within our study area, Brewer's and vesper sparrow were highly common sagebrush-associated songbirds with only a few occurrences ($< 1\%$ of total observations) of other sagebrush-associated species (e.g., sage thrasher [*Oreoscoptes montanus*], sage sparrow [*Artemisiospiza nevadensis*]).We built a suite of models using these nest density covariates, land enrollment in conservation program (SGI or non-SGI), Julian date, vegetation metrics, weather measures, and additive models amongst covariates (S3 and S4 Tables). We used the 'lme4' package in program R for our nest survival analysis. The top model(s) was selected using AIC$_c$.

## Adult density analysis

Using program R and JAGS [48], we used a multispecies dependent double-observer abundance model (MDAM) to estimate adult density for each species [37]. Adult density was estimated as a function of a species-specific intercept and fixed effects for SGI status and year. We also included a random effect for each plot to account for site-level variation not captured by covariates. We modeled detection as a function of observer, species, and year [37]. We used a vague normal distribution, N (0, 1000), for the priors of the linear predictor of mean species abundance per plot in each year and a prior uniform distribution ranging from 0 to 100 for the random plot effect [31, 37]. We used Markov chain Monte Carlo with three Markov chains consisting of 10,000 iterations. The first 1000 iterations were discarded as burn-in. We visually inspected trace plots, checked Rhat statistics, and examined the posterior density distributions to check for smooth, unimodal posterior distributions.

**Table 1. Brewer's and vesper sparrow nest densities by year and land enrollment in conservation program.** Estimates of nest density (number nests/25 ha) with 95% confidence intervals in parentheses, with nest and plot sample size listed for Brewer's sparrow and vesper sparrow in central Montana, USA from 2016–2018. SGI represents plots in pastures participating in the Sage Grouse Initiative conservation-based program, while non-SGI represents plots not enrolled.

| Species | Year/Enrollment | Nest Sample Size | Plot Sample Size | Estimated Plot Density | 95% CIs |
|---|---|---|---|---|---|
| Brewer's Sparrow | *SGI* | | | | |
| | 2016 | NA | NA | NA | NA |
| | 2017 | 10 | 5 | 4.19 | (2.84, 5.53) |
| | 2018 | 4 | 2 | 3.86 | (3.58, 4.14) |
| | **2016–2018** | **14** | **7** | **4.025** | **(3.21, 5.83)** |
| | *Non-SGI* | | | | |
| | 2016 | 18 | 6 | 6.01 | (4.20, 7.82) |
| | 2017 | 22 | 6 | 5.21 | (4.80, 5.61) |
| | 2018 | 12 | 4 | 5.59 | (4.80, 5.60) |
| | **2016–2018** | **52** | **16** | **5.60** | **(4.60, 6.34** |
| Vesper Sparrow | *SGI* | | | | |
| | 2016 | 11 | 5 | 5.64 | (1.95, 9.32) |
| | 2017 | 10 | 4 | 6.05 | (5.94, 6.16) |
| | 2018 | 15 | 6 | 7.13 | (6.53, 7.73) |
| | **2016–2018** | **36** | **15** | **6.27** | **(4.81, 7.69)** |
| | *Non-SGI* | | | | |
| | 2016 | 13 | 6 | 5.54 | (3.94, 7.14) |
| | 2017 | 16 | 6 | 5.96 | (5.59, 6.32) |
| | 2018 | 14 | 5 | 7.11 | (6.64, 7.57) |
| | **2016–2018** | **43** | **17** | **6.20** | **(5.393, 7.01)** |

## Comparing adult pair density and nest density

After estimating adult pair densities and nest densities separately, we tested for a linear relationship between the 2 estimates using a Pearson's correlation test. This approach allowed us to explore signals of density-dependent relationships. We assumed a 1:1 sex ratio given Brewer's and vesper sparrows are socially monogamous [57, 58] and that distinguishing between males and females in the field during adult surveys can be difficult without large time commitments observing individuals, pairs, or the use of auxiliary markers (e.g., transmitters, individually colored bands). We did not use singing behavior to determine the sex of each bird we observed, as female songbirds of some species have also been known to vocalize [59–61]. We assume that two individuals of the same species equal one adult pair. However, we acknowledge this assumption may be violated given that Brewer's sparrow and vesper sparrow are known to have multiple broods during a season.

## Results

### Nest density

From 2016–2018, we located 59 Brewer's sparrow nests and 65 vesper sparrow nests during nest transect surveys. While more nests were located opportunistically (*n* = 53, 86 respectively), our distance sampling methods did not allow the inclusion of nests found opportunistically in nest density analyses (e.g., lack of distance data because no schematic transect was being surveyed). We surveyed the same 80 plots each year, with 33 plots containing ≥ 2 Brewer's sparrow nests and 39 plots containing ≥ 2 vesper sparrow nests across all three years. Of the 80 plots surveyed each year, 21 plots for Brewer's sparrow had ≥ 2 nests every year of the study, and 24 plots had ≥ 2 vesper sparrow nests every year of the study. The remaining plots

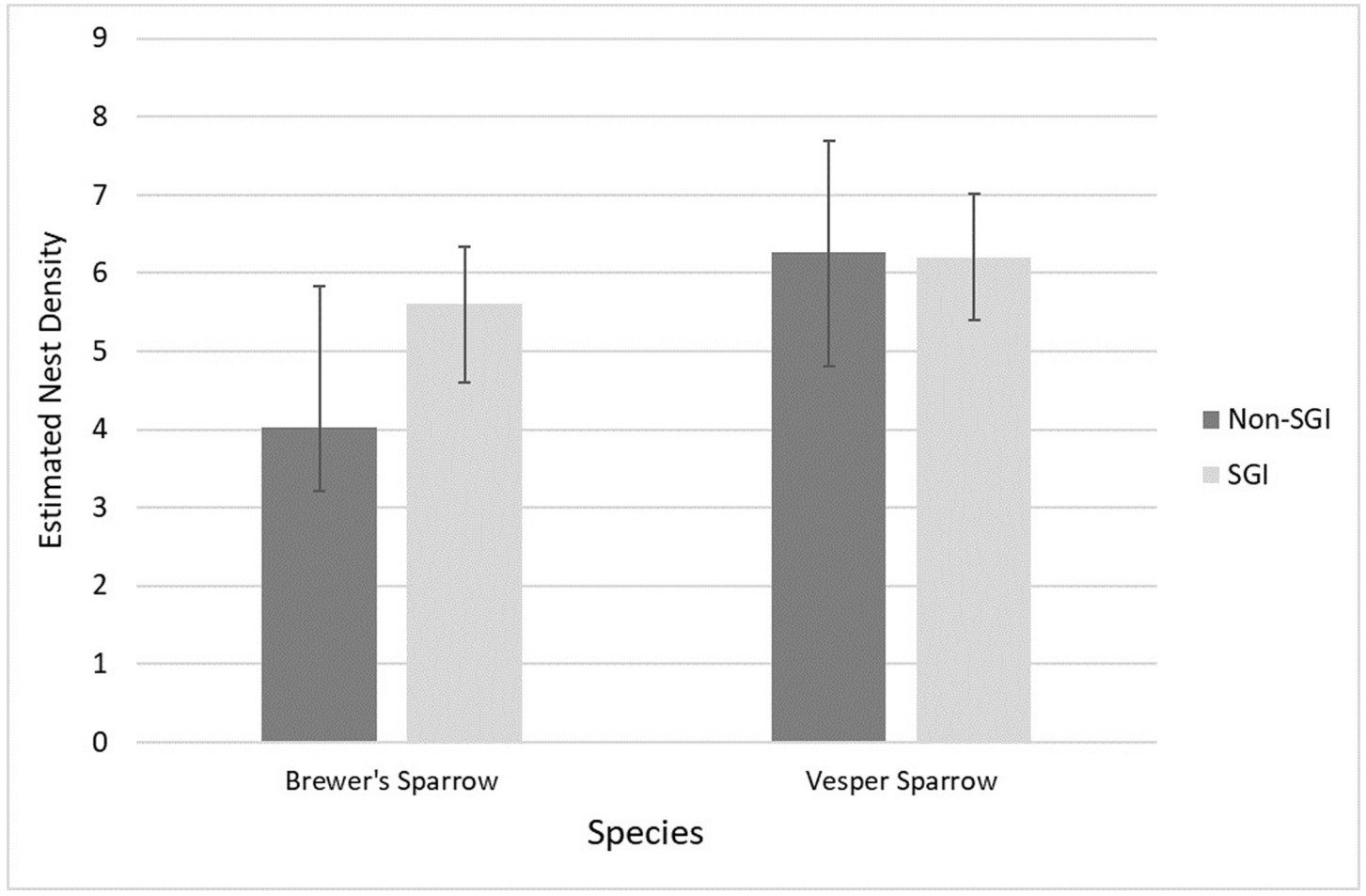

**Fig 1. Brewer's and vesper sparrow nest densities by grazing regime.** Plot reflecting estimates of nest density (number nests/25 ha) with 95% confidence intervals for Brewer's and vesper sparrow based on land enrollment from 2016–2018.

with ≥ 2 nests (12 with Brewer's sparrow nests and 15 with vesper sparrow nests) differed slightly among years for both species. The average detection probability was 0.38 ± 0.07 (mean ± standard error) for Brewer's sparrow nests and 0.38 ± 0.05 for vesper sparrow nests.

We estimated nest density by year and conservation program enrollment status of the plot (SGI or non-SGI). In 2016, we did not locate ≥2 Brewer's sparrow nests on any plots categorized as SGI. In 2017, Brewer's sparrow nest density estimates were slightly lower for SGI than non-SGI plots but had 95% overlapping confidence intervals. However, in 2018, Brewer's sparrow nest densities were lower on plots classified as SGI than non-SGI, and confidence intervals did not overlap, suggesting different bird responses on pastures participating in the partnership-based conservation program (Table 1). For vesper sparrows in all years, plots classified as SGI and non-SGI plots showed similar density estimates with overlapping confidence intervals at the plot level (Table 1), suggesting this species had no or minimal response to the partnership-based program. However, we also acknowledge our sample size limits the strength of our inference.

## Nest survival

We estimated DSR from 112 Brewer's sparrow nests and 151 vesper sparrow nests found both through distance sampling and opportunistically. Fifty-nine percent (66/112) of Brewer's sparrow nests and 49% (74/151) of vesper sparrow nests successfully fledged at least one offspring, for an average apparent nest survival rate of 53% (140/263) for both species. Predation was the most common cause of nest failure for both species, where predation accounted for 75% (35/46) of Brewer's sparrow nest failure and 81% (62/77) of vesper sparrow nest failure. Our top model for Brewer's sparrows (S4 Table) estimated seasonal nest survival to be 13.9%, with an average daily survival rate of 90% ($0.90^{19}$). Our top model for vesper sparrows (S4 Table) estimated seasonal nest survival to be 4.2% ($0.81^{15}$), with an average daily survival rate of 81%. Seasonal nest surivival was found by taking the average daily survival rate and multiplying it over the average nesting (egg laying to fledging) period. We used 19 days as the nesting period for Brewer's sparrow and 15 days as the nesting period for vesper sparrow.

The nestling stage had a higher DSR than the egg stage for both species each year of the study. The average DSR was highly variable between years, ranging from 0.61 ± 0.09 for vesper sparrows in 2018 to 0.93 ± 0.01 for Brewer's sparrows in 2016. The nest survival model for Brewer's sparrow that received the most support included shrub cover, and average maximum temperature, accounting for 89% of the relative support of the data ($\Delta AIC_c$ = 450.833, $w_i$ = 0.887, S4 Table). In the top model, nest survival increased with increased shrub cover (β = 0.101 ± 0.032) and decreased as maximum average temperature increased (β = −0.109 ± 0.040; Fig 2 and S4 Table). For vesper sparrows, the top model, which accounted for 71% of the relative support of the data ($\Delta AIC_c$ = 711.734, $w_i$ = 0.714, S4 Table) included slight positive effects of land enrollment in conservation program (β = 0.703 ± 0.158; Fig 3). The next best model accounted for 26% of the relative support of the data included both a slight positive effect of land enrollment (β = 0.703 ± 0.158) and a possible slight negative effect of species-specific nest density (β = -0.003 ± 0.0147).

## Adult pair density

Over the 3 study years, 9,156 adults were observed, including 4,119 observations of Brewer's sparrows and 5,037 vesper sparrows on the 80 plots (each plot 500 x 500 m [25 ha]) sampled each year. The average detection of Brewer's sparrows across years was 0.48 (95% Credible Interval [CI] 0.42–0.54) and the average detection of vesper sparrows was 0.47 (95% CI 0.41–0.54). Brewer's sparrow and vesper sparrow pair density estimates were highest in 2016 and lowest in 2018 (Fig 4). Plots enrolled in the conservation program had a negative effect on adult pair density for Brewer's sparrows (-0.10, 95% CI -0.9–-0.08) and no effect for vesper sparrows (-0.09, 95% CI-0.29–0.11).

## Adult pair density and nest density

At the plot-level, Brewer's sparrow nest density estimates were lower than their corresponding estimated adult pair densities for all but one plot. Pearson's correlation test between Brewer's sparrow nest density and adult pair density suggested a weak negative relationship ($r$ = −0.279, $p$ = 0.186), whereby the density of Brewer's sparrow pairs was 1.5–2 times higher than nests (Fig 4A). Conversely, the difference between vesper sparrow adult pair density and nest density was not significant ($r$ = 0.064, $p$ = 0.782). Thus, in contrast to Brewer's sparrow, the average density of vesper sparrow pairs at the plot-level was similar to the estimated number of nests (Fig 4B). The correlation did not change based on SGI status for either species.

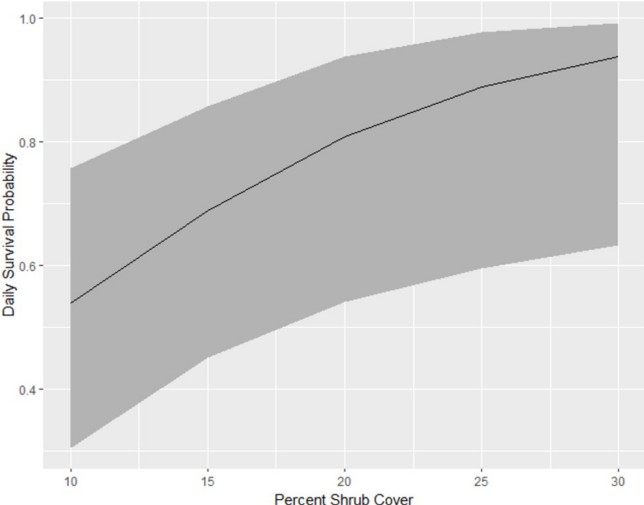

**Fig 2. Brewer's sparrow daily nest survival.** Daily nest survival probability and 95% confidence intervals (gray) for Brewer's sparrow nests as a function of percent shrub cover 2016–2018 in central Montana, USA.

## Discussion

Our study furthers information on density-dependent regulation in 2 sagebrush-steppe associated songbird populations by including nest density, as well as adult density and nest survival which has been explored in other studies [e.g., 16–18]. First, we investigated relationships between nest density and nest survival based on the 2 primary mechanisms of density-dependent regulation- habitat and behavior. We hypothesized that, given the lower shrub and forb cover of our study areas [32], nest survival was density-dependent, which would result in a negative effect of nest density on nest survival. Based on the lack of a relationship between nest density and nest survival for both Brewer's and vesper sparrows, our hypothesis was rejected. Land enrollment was the strongest predictor of vesper sparrow nest survival, where plots on land enrolled in the conservation program had slightly higher survival rates. The second best

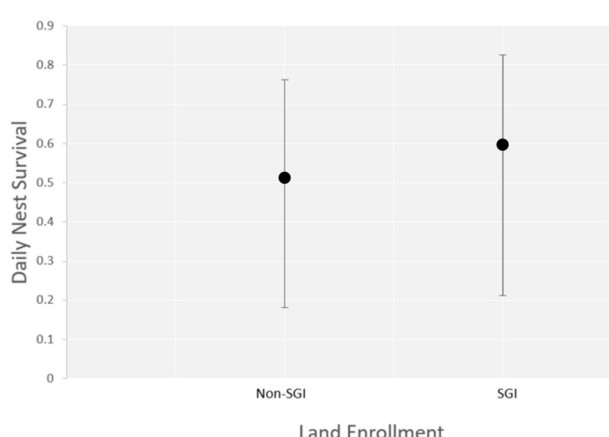

**Fig 3. Vesper sparrow daily nest survival.** Daily nest survival probabilities with 95% confidence intervals for vesper sparrow nests based on land enrollment from 2016–2018 in central Montana, USA.

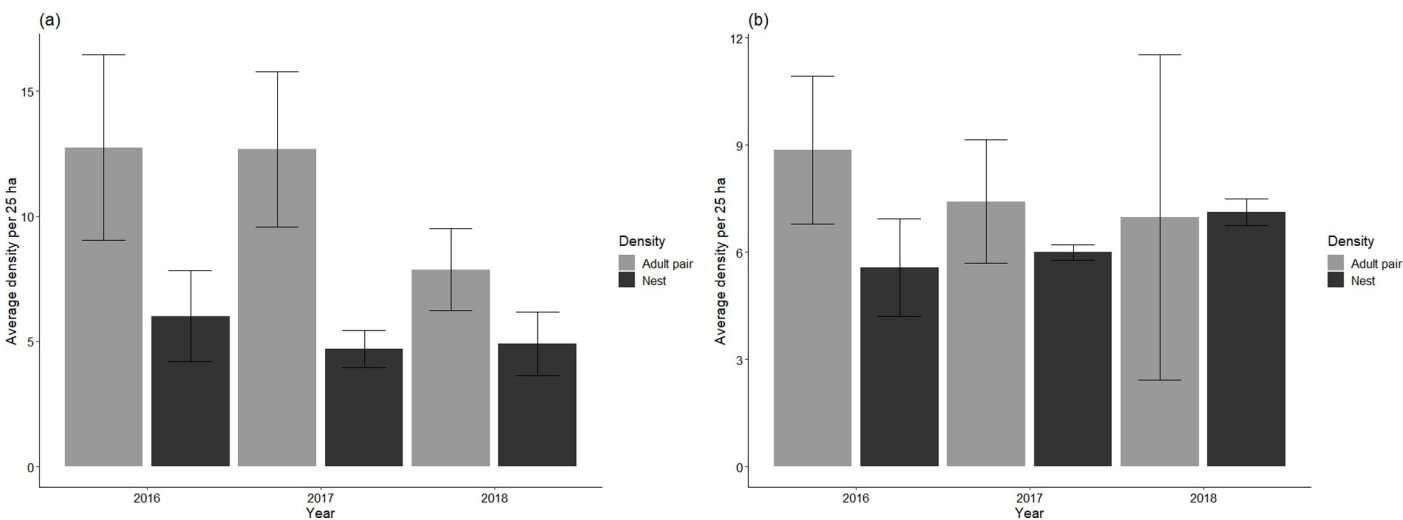

**Fig 4. Brewer's and vesper sparrow adult pair densities and nest densities.** Average estimated adult pair density and nest density at 25 ha plots with 95% confidence intervals for (a) Brewer's sparrows and (b) vesper sparrows from 2016–2018 in central Montana, USA.

model included an additive effect of species-specific nest density that was a very slight negative to no effect of species-specific nest density. Thus, density dependence may not be operating on nest survival within these populations or may be doing so differently than we examined.

Secondly, we explored the relationship between nest density and adult density in the 2 songbird populations. We rejected the hypothesis of a linear association between adult pair density and nest density for Brewer's sparrows but not vesper sparrows. Brewer's sparrow adult pair density was higher than nest densities across all plots (Fig 4A). Reynolds [62] observed that only 23% of Brewer's sparrow males initiated nests in southeastern Idaho, with the remaining males unsuccessful in defending territories or relocated to another area. Beyond intraspecific competition, interspecific competition can result in the lack of quality nest sites leading individuals to forego breeding for a season [63], leading to a greater density of adults compared to nests. However, vesper sparrow adult density and nest density were more similar than Brewer's sparrow (Fig 4), suggesting most adult vesper sparrows were nesting during the study. Breeding phenology of vesper sparrow is slightly earlier than Brewer's sparrow in our study area (Ruth pers.comm). Thus, vesper sparrows may be out competing Brewer's sparrows for nesting sites. Further studies should explore if interspecific competition is a mechanism of density-dependence between the two species. Lastly, we acknowledge we lack information on 1) the amount of multiple brooding efforts per year for the two species and 2) the relationship between individual nesting territories and the sample plot. For instance, vesper sparrows have expanded territories in which they can travel distances greater than our plot size ($> 500$ m$^2$) for activities such as foraging [64]. Incorporating information on brooding attempts and nesting territories is needed to further understand the relationship between adult pair density and nest density. However, from a conservation perspective, we emphasize non-breeders or migrants may exist in songbird populations; thus, the number of adults does not represent breeding activity, particularly for Brewer's sparrows as suggested in our study.

Knowing the factors or mechanisms affecting site-specific nest density, nest survival, and adult density can inform the processes regulating populations. Nest density for both Brewer's and vesper sparrows varied slightly across years and conservation program enrollment status. Brewer's sparrows are sagebrush obligates that nest in sagebrush shrubs, while vesper sparrows nest at the base of sagebrush shrubs or alongside grass [64, 65]. The exact mechanisms that

caused the differences in the estimates for Brewer's sparrow nest densities between plots enrolled (SGI) or not enrolled (Non-SGI) in the conservation program but not in the estimates for vesper sparrows are unclear. However, the confidence intervals between SGI and Non-SGI are overlapping suggesting no to minimal differences. Domestic livestock grazing primarily affects grass and ground vegetation structure [32]. Nest density and nest success did not differ based on conservation program status for thick-billed longspurs, a grassland-associated species, in our study area [66]. Similarly, greater sage-grouse nest success in our study area was not affected by differences in the SGI status of pasture [32]. Additionally, metrics of vegetation covariates used in our analyses did not differ based on land enrollment (S2 Table).

Further, while the conservation partnership-based program's ecological interest in our study area is to improve habitat for sage-grouse using domestic livestock grazing as the management tool, the program is designed to have the flexibility to support private landowners' socioeconomic concerns. For instance, landowners could adjust stocking rates based on pasture conditions (e.g., vegetation characteristics mediated by precipitation events, infrastructure demands such as stock tanks). Thus, the implementation of specific grazing regimes on plots in pastures participating in the program were dynamic during our study, making it difficult to isolate grazing differences (e.g., ranch specific differences, timing of grazing relative to nesting period, grazing intensity). In addition, information on stocking rates or more detailed grazing intensity metrics (e.g., cattle pat counts; [32]) are lacking.

Nest survival models for both species suggested annual variation in DSR and suggest highly variable nest survival for both species. Brewer's sparrow average annual DSR values ranged from 0.79–0.94, which are within reported estimate ranges from across its distribution [40, 67]. Similarly, the average annual vesper sparrow DSR ranged from 0.52–0.79, which aligns with observations in previous studies [68, 69]. Brewer's sparrow DSR increased with shrub cover, emphasizing the importance of shrubs to this sagebrush-obligate songbird. Annual variation in precipitation may explain annual variation in Brewer's sparrow DSR. For example, Brewer's sparrow nest survival rates were lowest in 2017, the year with the lowest average monthly precipitation [33]. Precipitation and other abiotic factors have been suggested to play important roles in songbird demographics during the breeding season [70, 71].

While we did not find direct effects of conservation program on Brewer's and vesper sparrow demographics, we caution against suggesting that the partnership-based program is irrelevant. The conservation program engages private landowners in conservation activities, instilling a community-based effort. From an ecological perspective, conservation-based program activities may not be a "one size fits all" solution [31], but broader advantages, such as preventing habitat loss, should be recognized. Lastly, detecting and understanding possible population regulation mechanisms across demographic parameters can reveal which species need more information and if only measuring adult density may be appropriate. Based on our findings, monitoring vesper sparrow populations through adult pair density would be appropriate, as nest densities were similar to adult pair densities (Fig 4). In contrast, Brewer's sparrow adult pair density did not reflect nest density. Thus, monitoring nesting parameters, such as nest density and nest survival, may be more relevant for this species. We encourage future studies and monitoring programs to consider the patterns amongst multiple songbirds demographic parameters when informing conservation agendas.

## Supporting information

**S1 Table. List of covariates used in estimating nest density, nest survival and adult density.** (DOCX)

**S2 Table. Comparison of biotic and abiotic covariates used based on land enrollment.**
(DOCX)

**S3 Table. Distance sampling model results for Brewer's sparrow and vesper sparrow.**
Model selection table for detection rates of Brewer's and vesper sparrow nests using distance sampling.
(DOCX)

**S4 Table. Nest survival model results for Brewer's sparrow and vesper sparrow.** Model selection table for Brewer's and vesper sparrow nest survival. Top model set is shown with different Akaike's Information Criterion ($\Delta AIC_c$) ranked by descending model weight.
(DOCX)

**S1 Fig. Schematic of the dependent double-observer avian survey method.** The primary (open circle) and secondary observer (dashed circle) walk single file along the transect (dotted line) within a 500 m x 500 m sampling plot. Observers survey up to 125 m on either side of the transect (dotted line). All surveys start at the lower right corner of the sample plot. Red arrows indicate the direction of travel.
(DOCX)

**S1 Data.**
(CSV)

**S2 Data.**
(CSV)

**S3 Data.**
(CSV)

## Acknowledgments

We thank local private landowners and ranchers for allowing access to their land to collect data, and S. K. Bundick, A. S. Emmel, G. Fiske-White, E. D. Green, J. D. Golding, A. H. Harrington, J. L. Harris, K. M. Keohane, E. Kern-Lovick, J. Lee, K. Macco-Webster, P. M. Newberry, V. T. Nguyen, K. M. Pacher, J. Pierce, A. C. Richette, K. M. Reinstma, M. D. Scheuering, C. W. Stephens, S. L. Stevens, L. A. Sutcliffe, T. J. Swartout, K. Weeks, and V. Vidal-Astudillo for their assistance surveying adult songbirds and locating and monitoring nests. We are extremely grateful to M. Szczypinski for his invaluable assistance during field data collection. Additionally, we would like to thank the three anonymous reviewers for their thoughtful critiques and recommendations.

## Author Contributions

**Conceptualization:** Kayla A. Ruth, Victoria J. Dreitz.

**Data curation:** Kayla A. Ruth, Kaitlyn M. Strickfaden.

**Formal analysis:** Kayla A. Ruth, Kaitlyn M. Strickfaden.

**Funding acquisition:** Lorelle I. Berkeley, Victoria J. Dreitz.

**Investigation:** Kayla A. Ruth.

**Methodology:** Kaitlyn M. Strickfaden, Victoria J. Dreitz.

**Project administration:** Lorelle I. Berkeley, Victoria J. Dreitz.

**Resources:** Victoria J. Dreitz.

**Supervision:** Victoria J. Dreitz.

**Writing – original draft:** Kayla A. Ruth, Kaitlyn M. Strickfaden.

**Writing – review & editing:** Kayla A. Ruth, Lorelle I. Berkeley, Kaitlyn M. Strickfaden, Victoria J. Dreitz.

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
