## [Decision Letter · Decision Letter 0]

22 Sep 2022

PONE-D-22-21628Density dependence of songbird demographics in grazed sagebrush steppePLOS ONE

Dear Dr. Ruth,

Thank you for submitting your manuscript to PLOS ONE. After careful consideration, we feel that it has merit but does not fully meet PLOS ONE’s publication criteria as it currently stands. Therefore, we invite you to submit a revised version of the manuscript that addresses the points raised during the review process.

This paper examines multiple measures of the demography on songbirds and the potential impacts of grazing.  It fills a knowledge gap that is important for conservation and management, but the paper needs substantive improvements to be suitable for publication. 

The study is described clearly and makes a compelling argument for considering density dependence when evaluating songbird populations.  However, there are some key improvements needed in the framing, analysis, presentation of results and the discussion.

Both reviewers identify a number of ways that the framing of the paper and the introduction, in particular could be improved.  As Reviewer #1 indicates, the paper is attempting to address two important aspects but the focus of the introduction is primarily on only one.  Reviewer #2 also highlights important contextual information that is missing to understand the potential effects of grazing or other potential factors (e.g., predation upon nests).  It might be useful to include some additional information in the study area description section.  Both reviewers indicate that there is a need for more ecological context throughout the introduction and discussion. For example, as Reviewer # 2 points out: if predation is the major source of nest mortality, why did you anticipate that grazing would be a major influence on nest success? There is little to no discussion of habitat quality other than a description of preference for nest siting. Both reviewers mention a number of alternative explanations that should be considered for many of the results.  Even if this is outside the scope of the study, there should be an acknowledgment of these other factors or potential explanations.

The paper would be strengthened if there were improvements in the analysis and interpretation of the results that includes more clear statement of the assumptions and/or limitations.  There are a number of conclusions, pointed out by both reviewers, that do not seem to be directly attributable to your results (e.g., birds forgoing breeding) that require some other type of support.  The reviewers also make some good suggestions as to how to improve the presentation and interpretation of the results. There are detailed suggestions by both reviewers that will improve the manuscript, reframe the paper to make it better highlight the strengths, and increase relevance of the conclusions.  These are substantive and will need to be addressed.  Some additional suggestion are below. With substantial improvements this paper could make a welcome contribution to the field.

Suggestions:

In Figure 4, if the desire is to emphasize the difference between the two types of grazing plots, consider rearranging these graphs into (a) adult density per year per plot type and (b) nest density per year per plot to facilitate the comparison.A set of box and whisker plots could include most of the results shown in Table 1 and could be arranged to either emphasize the comparison of species within plots or across plots, depending on which you prefer to highlight.Lines 243-245.  Sentence should read: …but had overlapping 95% confidence intervals.Lines 351-355.  Could total nest density be an indicator of habitat quality and less about predator detection/saturation or some combination? Can you deduce habitat quality patterns from nest success data or nest density?Lines 389-391: Sentence should read:  …can reveal for which species more information is needed and for which only measuring…==============================

We look forward to receiving your revised manuscript.

Kind regards,

Karen Root, Ph.D.

Academic Editor

PLOS ONE

Journal Requirements:

"Funding for this project was provided by the sale of hunting and fishing licenses in Montana, Federal Aid in Wildlife Restoration Grant F16AF00294 (MT #W-165-R-1) to Montana Fish Wildlife and Parks; Montana Fish, Wildlife and Parks; the Montana Fish, Wildlife and Parks Tax Checkoff Program; Montana Cooperative Wildlife Research Unit; and Safari Club International Foundation, Montana."

 "LIB - Federal Aid in Wildlife Restoration Grant F16AF00294 to Montana Fish Wildlife

and Parks. VJD - Montana Fish Wildlife and Parks awarded grant W-158-R-1 to University of

Montana fwp.mt.gov

www.fws.gov/program/wildlife-restoration

The funders had no role in study design, data collection and analysis, decision to

publish, or preparation of the manuscript."

4. We note that Figure 1 in your submission contain map images which may be copyrighted. All PLOS content is published under the Creative Commons Attribution License (CC BY 4.0), which means that the manuscript, images, and Supporting Information files will be freely available online, and any third party is permitted to access, download, copy, distribute, and use these materials in any way, even commercially, with proper attribution. For these reasons, we cannot publish previously copyrighted maps or satellite images created using proprietary data, such as Google software (Google Maps, Street View, and Earth). For more information, see our copyright guidelines: http://journals.plos.org/plosone/s/licenses-and-copyright.

a) You may seek permission from the original copyright holder of Figure(s) [#] to publish the content specifically under the CC BY 4.0 license.  

Reviewers' comments:

Reviewer's Responses to Questions

**Comments to the Author**

1. Is the manuscript technically sound, and do the data support the conclusions?

Reviewer #1: Partly

Reviewer #2: Partly

2. Has the statistical analysis been performed appropriately and rigorously? 

Reviewer #1: Yes

Reviewer #2: I Don't Know

3. Have the authors made all data underlying the findings in their manuscript fully available?

Reviewer #1: Yes

Reviewer #2: Yes

4. Is the manuscript presented in an intelligible fashion and written in standard English?

Reviewer #1: Yes

Reviewer #2: Yes

5. Review Comments to the Author

Reviewer #1: This manuscript compares nest and adult densities to nest survival for two songbird species in the sagebrush steppe. Given declining songbird populations and the numerous threats facing the sagebrush ecosystem, this paper represents an important contribution to the literature. However, I have some concerns, most of which may be able to be addressed through clarification and restructuring of the manuscript.

My main concern is that the focus seems somewhat confused throughout the paper. The two main points (as I understood them) were the ideas of density dependence and grazing effects. However, the paper seemed to bounce back and forth between the two ideas without really connecting them or clarifying why they should be addressed together. This makes it difficult to identify the key takeaway points. I think this could be addressed through some restructuring and further explanation, but the intro seems to focus on grazing effects while much of the rest of the paper is devoted to ideas of density dependence.

My second concern is regarding the grazing effects. I’m not convinced that you have enough information to actually test the effects of grazing. As described, you’ve only separated plots into SGI and non-SGI but there is no information on stocking rates, specific timing of grazing, duration of grazing, grazing intensity etc., in either of the systems. You state that non-SGI ranches were “generally managed less intensively”, but you also suggest that there was variation within each group (e.g., SGI was tailored to the specific ranch) and without more detail there is no reason to think that differences within a group wouldn’t be equal or greater to differences between groups. Furthermore, plots within the SGI group could be grazed at any point within the rotation and it would be reasonable to expect effects of grazing to differ if a specific plot was grazed in May vs. November (e.g., when vegetation is removed relative to the nesting period). Without any other habitat factors being considered, it is hard to isolate grazing regime as the driving factor behind any potential differences. Without more detailed information on grazing systems, it isn’t possible to judge whether this broad categorization of SGI vs. non-SGI is meaningful. Given this uncertainty, I think the emphasis on the effects of grazing, particularly in the intro, is unwarranted and overstates the reach of this paper.

Specific comments

Line 31: The abstract starts with livestock grazing which, given the rest of the paper and particularly the discussion, seems to be overstating the problem, especially because not much of the discussion focuses on grazing.

Line 44: The abstract started big focusing on evaluating the effects of grazing but, as written, this result seems tacked onto the main focus of the paper (density-dependence).

Line 55: missing closing parentheses

Line 56: While I certainly don’t disagree with the statement, it is a very broad claim with no citations or explanation, and it seems added onto this paragraph as an afterthought rather than being developed as its own idea.

Line 74: Again, no citation, although I can think of several studies that have looked at grazing effects on songbirds.

Line 75: This entire paragraph doesn’t have any citations, even though the authors make some broad claims.

Line 88: This is a confusing switch from talking about density-dependence to the relationship between adult and nest densities. It isn’t clear if you’re focusing on the relationship between adult and nest densities from the angle of density dependence (more theoretically) or management (e.g., do we only care about this relationship because it could mean we could just count adults and not monitor nests), so this could be better explained and linked together.

Line 90: I think it would be helpful to clarify why we care about density dependence. The authors seem to be dancing around the issue, but it isn’t clear why, particularly from a management perspective which is how the authors have framed the paper, understanding mechanisms of density dependence is important

Line 94: Here you’re bouncing back and forth between idea of grazing (started with this broadly and now have come back to it) and density dependence, which is confusing. As written, you’re just interested in how grazing affects bird densities and survival, not how it influences density dependence (which may be the case but is confusing since you had switched to the idea of density dependence).

Line 107: These seem like different objectives kind of cobbled together.

Line 109: This is the first mention of trying to compare effects of grazing and density dependence (which I think is what you’re trying to say?). That’s a curve ball to throw into your objectives.

Line 133: I’m not convinced that you have enough information on grazing regime to be able to differentiate these two or that there is a meaningful difference that would allow you to say anything about the effects of grazing. There’s no information on timing of grazing, grazing intensity or duration, or stocking rates.

Line 137: What other biotic factors did you examine? Reading further, it isn’t clear that you incorporated any additional variables beyond grazing regime, year, and date. The lack of any habitat variables seems like a problem, particularly for nest survival.

Line 171: Do you mean 95% of birds would be observed (and not missed) using this method? As written, it is a little confusing.

Line 182: Maybe I missed it, but did you look at effects of anything (habitat, grazing, etc) on nest density? That isn’t clear from this section. You say you used AIC to select among competing models, but don’t explain what those are.

Line 199: I assume you mean you included a random effect for ‘plot’? Not completely clear.

Line 303: Maybe I misunderstood, but I thought you were testing the correlation between adult and nest densities. These results would suggest that the correlation was not significant for either species. If there is no correlation, how does that translate to the number of pairs being the same as the number of nests? Either this needs to be clarified or corrected.

Line 307: Given all the issues with apparent survival estimates, how is this useful?

Line 330: Here the lack of an effect could also be because you didn’t measure or compare meaningful differences in grazing regime (i.e., that your two grazing categories are not meaningfully different).

Line 337: Any anecdotal mention of how these years differed? Any significant differences in precipitation across the 3 study years?

Line 342: These appear to be two competing explanations, but there’s no connection/flow between them in this paragraph.

Lines 353-355: Here you start by arguing that adults may be less vigilant if other birds are around, by which I think you’re implying that they could be more successful because they don’t have to use as much of their energy reserves being vigilant? However, the next sentence talks about prey saturation being the driving factor for reduced predation. These seem to be different (although not mutually exclusive) ideas, but the writing (“In this way. . .”) seems to connect the two ideas and neither seems to be explored in enough detail.

Line 376: This is the first mention of this and seems like a concern that should be raised and explained in your methods.

Line 392: Again, your results suggest that there was no correlation for either species.

Line 396: I think you could strengthen your manuscript by more (and earlier) discussion about why we care about density dependence. You seem to be arguing from a management perspective, but the links haven’t been fully developed or explained. Some additional explanation would help pull that together and help the reader understand why we should care about this paper.

Reviewer #2: Comments to Authors, PONE-D-22-21628

General Remarks:

Note: I also reviewed a previous submission of this manuscript for a different journal. The previous and current versions of the manuscript are very similar, as are my comments and concerns. I admittedly found it disconcerting that the authors apparently elected not to incorporate most suggestions or address the concerns articulated in the previous round of reviews, as peer reviews necessitate considerable effort and time on the part of reviewers. Despite the similarities between versions, however, I re-read and evaluated the current version thoroughly.

The authors of this paper assessed relationships between livestock grazing regimes and songbird demography within a habitat type that has been altered extensively range-wide. Moreover, they attempted to assess relationships between adult density, nest density, and nest survival. I concur that understanding of grazing effects on songbirds and the extent to which density-dependence operates on nest survival (and within which contexts) is lacking. Moreover, understanding the relationship between the number of adults within an area and actual reproductive effort (herein, nest density) is important for assessing how meaningful density is as a proxy within different systems. As such, the authors have identified some important gaps in understanding, both broadly across species, and within the focal system from a conservation perspective.

I also found the writing to be generally clear and concise.

Whereas the spatial replication (n = 80, 25-ha study plots) and study duration (n = 3 years) were impressive, I still have some strong reservations about whether the study approach can suitably address the objectives related to density, and whether the system was a logical one within which to expect and evaluate density-dependence. The study took place within an area that had very low densities of the two focal species compared with other parts of their range. I also think that the authors put forth some explanations (e.g., pairs skipping breeding during some years) that are unwarranted given the approach and data, and could be unnecessarily misleading to readers in terms of furthering understanding of the species. I elaborate on these concerns, and provide some additional suggestions, below.

Specific Comments:

With respect to the tests of grazing effects, I did not find any description of the extent of grazing pressure (e.g., average number of cattle per ha or similar). Without such information, readers cannot compare grazing levels across spatiotemporal contexts. Even providing a range of typical livestock densities present in the study plots would be helpful. Without this information, readers cannot assess whether the grazing regimes were comparatively low, moderate or high. Perhaps grazing was not influential simply because livestock densities were low enough to have minimal influence on habitat metrics that could influence food or nest predation rates for breeding songbirds, whereas at higher grazing levels negative effects could result.

Indeed, the introduction emphasizes habitat changes in the sagebrush steppe, and the potential influence of grazing regimes, whereas the grazing aspect seems sort of “tacked on” when it comes to the articulation of the study questions at the end of the introduction.

I also found myself craving more information about the habitat structure/composition of the SGI versus non-SGI plots. Being able to more explicitly link the two grazing regimes to potential differences in resultant habitat characteristics and then those to avian responses would be more compelling than simply comparing the effects of the two grazing treatments (especially without the inclusion of grazing intensity). This would be especially helpful for understanding why “grazing regime had a negative effect on adult pair density for Brewer’s sparrows” (lines 289 – 290).

I think more ecological context would strengthen the premise of the study. For example, the primary source of nesting mortality was nest predation. How/why did you anticipate that grazing in your system could affect egg and/or nestling predation rates (lines 95 – 96)? Do you have a sense for the primary nest predator species?

The sample sizes of nests (N = 59 BRSP and 65 VESP) located via transect surveys were extremely low relative to other studies, especially considering the timespan (3 years) and n = 80 plots. This further exemplifies my confusion as to why the authors envisioned that their system was a logical one in which to expect density-dependent demographic responses. Certainly, local habitat quality influences the potential of an area to support individuals and the point at which density-dependence would manifest, but the authors do not describe key attributes such as mean/variation of shrub cover and height for comparison, other than to say in the Study area section that the area consists of “less shrub and forb cover and shorter shrubs than other sagebrush steppe areas”. I think at a minimum, in the discussion the authors should address their densities/nest sample sizes compared with other studies that have monitored BRSP and VESP nests, and more thoroughly address the types of systems in which density-dependent nest survival is most likely to operate.

Nest survival analysis section: I do not fully understand from the explanation provided (or supplementary data) how the two nest density covariates were calculated, with which to assess the effect of nest density on nest survival. Please clarify. Were the covariates that you used the distance to the nearest nests of each species? If so, then would this not more likely be reflective of spatial autocorrelation in rates of nest predation rather than nest densities per se?

I did not find figure 1 to be particularly necessary, as it does not convey useful spatial information beyond that described in the text.

I suggest standardizing your density data in figures to per square ha to be more directly comparable to other studies.

Because ratios of the densities of adults to nests (especially for BRSP) were skewed towards adults, the authors concluded (1) that some birds may forgo breeding within some years, and (2) evidence of density-dependence. I have many reservations about these conclusions, as fleshed out in the following bullets:

• There are many reasons in my estimation for why the number of nests observed in an area should be lower than the number of adults present, as not every pair has an active nest at a given point in time. Indeed, in this study, approximately half of nests failed, often during the incubation period, meaning that at any given time, some parents within an area would be between nesting attempts. Both species also are multi-brooded, so some parents would have been between nesting attempts even after the first brood fledged.

• Second, early-stage (build, pre-lay and lay) nests, during which adults often are not present, likely would not have been detected using the nest-searching method employed, which would bias nest numbers low. Both species will re-nest multiple times following nest failures, so early-stage nests exist throughout much of the season.

• The assumption of a 1:1 sex ratio for observed adults also is potentially problematic, because detection probabilities of males (who sing) versus females (more cryptic) can be very different which could bias the number of actual pairs high.

• The timing of relating the number of adults to nests within an area also matters. Whereas the authors indicated that surveys for adults and nest-searching were somewhat temporally matched, it was unclear at the analytical stage whether the authors compared the density of birds observed with nest densities within the same time period (e.g., week) or averaged across the season. Pairs may not have had an active nest at any given time for reasons described above. Summarizing the data across the entire season could therefore introduce considerable noise into the analyses.

• The concept that BRSPs would skip breeding within a year does not make sense for a relatively short-lived bird. Certainly, some males pair later than others, and some may not pair at all, especially if they are not able to procure and defend a suitable territory. Without monitoring individually-marked birds, however, one cannot discern whether particular males or females paired or initiated a nest at any point within a season or whether individuals whose first nests failed tended to disperse within a season to try to breed elsewhere. The authors cite one paper from forty years ago to support the idea of a large proportion of male BRSP not initiating nests within a season, but those authors similarly did not mark individual birds. Suggesting that the focal species “may not breed” every year as a tactic could therefore be very misleading.

6. PLOS authors have the option to publish the peer review history of their article (what does this mean?). If published, this will include your full peer review and any attached files.

Reviewer #1: No

Reviewer #2: No

---

## [Author Response · Author response to Decision Letter 0]

21 Nov 2022

Comments to the Author

Reviewer #1: This manuscript compares nest and adult densities to nest survival for two songbird species in the sagebrush steppe. Given declining songbird populations and the numerous threats facing the sagebrush ecosystem, this paper represents an important contribution to the literature. However, I have some concerns, most of which may be able to be addressed through clarification and restructuring of the manuscript.

My main concern is that the focus seems somewhat confused throughout the paper. The two main points (as I understood them) were the ideas of density dependence and grazing effects. However, the paper seemed to bounce back and forth between the two ideas without really connecting them or clarifying why they should be addressed together. This makes it difficult to identify the key takeaway points. I think this could be addressed through some restructuring and further explanation, but the intro seems to focus on grazing effects while much of the rest of the paper is devoted to ideas of density dependence.

My second concern is regarding the grazing effects. I’m not convinced that you have enough information to actually test the effects of grazing. As described, you’ve only separated plots into SGI and non-SGI but there is no information on stocking rates, specific timing of grazing, duration of grazing, grazing intensity etc., in either of the systems. You state that non-SGI ranches were “generally managed less intensively”, but you also suggest that there was variation within each group (e.g., SGI was tailored to the specific ranch) and without more detail there is no reason to think that differences within a group wouldn’t be equal or greater to differences between groups. Furthermore, plots within the SGI group could be grazed at any point within the rotation and it would be reasonable to expect effects of grazing to differ if a specific plot was grazed in May vs. November (e.g., when vegetation is removed relative to the nesting period). Without any other habitat factors being considered, it is hard to isolate grazing regime as the driving factor behind any potential differences. Without more detailed information on grazing systems, it isn’t possible to judge whether this broad categorization of SGI vs. non-SGI is meaningful. Given this uncertainty, I think the emphasis on the effects of grazing, particularly in the intro, is unwarranted and overstates the reach of this paper.

We concur with the Reviewer in discussing two very separate aspects – density-dependence and grazing –which were presented in a highly confusing manner and made the manuscript hard to follow. We have reorganized the manuscript substantially, especially in the Introduction and Discussion. Our primary interest is density-dependence, and we changed the manuscript to provide that focus.

In addition, we attempted to take advantage of a conservation program, the Sage Grouse Initiative, occurring in our study area. While the primary activity of this conservation program was grazing, there were many aspects of the program beyond the grazing implementation (e.g., infrastructure improvements – fences and stock tanks; private landowner engagement; economic values). In fact, during our study, we were not fully aware of when the grazing implementation occurred in the pastures of our sampling plots. We only conducted our sampling during the breeding season and suspect that some of our sample plots had grazing during seasons we were not present (e.g., fall and winter). Unfortunately, we did not collect information on grazing metrics during our study for several reasons. And we can not go back and change that at this time. Thus, we have changed the manuscript to reflect a ‘partnership-based conservation program’ and not a ‘grazing management program’ in which we explore if there is a difference in bird response based on the plot being in a pasture participating in the partnership-based conservation program.

Specific comments

Line 31: The abstract starts with livestock grazing which, given the rest of the paper and particularly the discussion, seems to be overstating the problem, especially because not much of the discussion focuses on grazing.

The abstract has been reworked based on reviewer comments. 

Line 44: The abstract started big focusing on evaluating the effects of grazing but, as written, this result seems tacked onto the main focus of the paper (density-dependence).

The abstract has been reworked based on reviewer comments. 

Line 55: missing closing parentheses

This has been fixed. 

Line 56: While I certainly don’t disagree with the statement, it is a very broad claim with no citations or explanation, and it seems added onto this paragraph as an afterthought rather than being developed as its own idea.

The introduction was reworked based on reviewer comments. 

Line 74: Again, no citation, although I can think of several studies that have looked at grazing effects on songbirds.

The introduction was reworked based on reviewer comments.

Line 75: This entire paragraph doesn’t have any citations, even though the authors make some broad claims.

The introduction was reworked based on reviewer comments.

Line 88: This is a confusing switch from talking about density-dependence to the relationship between adult and nest densities. It isn’t clear if you’re focusing on the relationship between adult and nest densities from the angle of density dependence (more theoretically) or management (e.g., do we only care about this relationship because it could mean we could just count adults and not monitor nests), so this could be better explained and linked together.

The introduction was reworked based on reviewer comments.

Line 90: I think it would be helpful to clarify why we care about density dependence. The authors seem to be dancing around the issue, but it isn’t clear why, particularly from a management perspective which is how the authors have framed the paper, understanding mechanisms of density dependence is important

We have substantially re-organized the Introduction, including providing an example, that we feel addresses the comment.

Line 94: Here you’re bouncing back and forth between idea of grazing (started with this broadly and now have come back to it) and density dependence, which is confusing. As written, you’re just interested in how grazing affects bird densities and survival, not how it influences density dependence (which may be the case but is confusing since you had switched to the idea of density dependence).

We recognize reviewer comments and tried to make necessary adjustments to bring together these different aspects of both ideas.

Line 107: These seem like different objectives kind of cobbled together.

We recognize reviewer comments and tried to make necessary adjustments to bring together these different aspects of both ideas.

Line 109: This is the first mention of trying to compare effects of grazing and density dependence (which I think is what you’re trying to say?). That’s a curve ball to throw into your objectives.

We recognize reviewer comments and tried to make necessary adjustments to bring together these different aspects of both ideas. 

Line 133: I’m not convinced that you have enough information on grazing regime to be able to differentiate these two or that there is a meaningful difference that would allow you to say anything about the effects of grazing. There’s no information on timing of grazing, grazing intensity or duration, or stocking rates.

Unforunately, we were not able to aquire more specific information or grazing metrics. We recognize that these metrics would have been useful in helping tease apart grazing effects of different demographic rates. 

Line 137: What other biotic factors did you examine? Reading further, it isn’t clear that you incorporated any additional variables beyond grazing regime, year, and date. The lack of any habitat variables seems like a problem, particularly for nest survival.

We recognize that additional biotic factors that were not collected may be influencing nest survival. 

Line 171: Do you mean 95% of birds would be observed (and not missed) using this method? As written, it is a little confusing.

This has been edited for clarity. 

Line 182: Maybe I missed it, but did you look at effects of anything (habitat, grazing, etc) on nest density? That isn’t clear from this section. You say you used AIC to select among competing models, but don’t explain what those are.

AIC was used to determine which key functions and adjustment terms were the best fit for the distance sampling data. 

Line 199: I assume you mean you included a random effect for ‘plot’? Not completely clear.

This has been clarified in the section “Nest Survival Analysis”. 

Line 303: Maybe I misunderstood, but I thought you were testing the correlation between adult and nest densities. These results would suggest that the correlation was not significant for either species. If there is no correlation, how does that translate to the number of pairs being the same as the number of nests? Either this needs to be clarified or corrected.

We concur with the reviewer about the presentation on this relationship. While the p-value suggested the relationship between nest density and adult density was not significant for either species, we believe more than a p-value needs to be considered when interpreting results from a statistical test. First, our sample size was relatively small for a correlation test. However, we felt this was the best type of statistical test to consider. Thus, why we explored the relationship between ratio of nests/adult pair and nest survival. We used the relationship also in our evaluation. 

Lastly, we changed our text throughout the manuscript to state this is a pattern that may provide some information into the processes regulating density-dependence.

Line 307: Given all the issues with apparent survival estimates, how is this useful?

At the plot level, we were unable to reliably estimate nest survival estimates, due to low sample sizes. Thus, apparent nest survival was used in order to show the relationship between nest survival and nest density per adult density at the plot level. 

Line 330: Here the lack of an effect could also be because you didn’t measure or compare meaningful differences in grazing regime (i.e., that your two grazing categories are not meaningfully different).

We have included further text acknowledging possible reasons why we had a lack of effect. 

Line 337: Any anecdotal mention of how these years differed? Any significant differences in precipitation across the 3 study years?

Yes, this is mentioned in lines 345-348. 

Line 342: These appear to be two competing explanations, but there’s no connection/flow between them in this paragraph.

We recognize reviewer comments and tried to make necessary adjustments to bring together these different aspects of both ideas. 

Lines 353-355: Here you start by arguing that adults may be less vigilant if other birds are around, by which I think you’re implying that they could be more successful because they don’t have to use as much of their energy reserves being vigilant? However, the next sentence talks about prey saturation being the driving factor for reduced predation. These seem to be different (although not mutually exclusive) ideas, but the writing (“In this way. . .”) seems to connect the two ideas and neither seems to be explored in enough detail.

We recognize reviewer comments and tried to make necessary adjustments to bring together these different aspects of both ideas and to aid in clarity. 

Line 376: This is the first mention of this and seems like a concern that should be raised and explained in your methods.

This concept has been added to the text. 

Line 392: Again, your results suggest that there was no correlation for either species.

We concur that the relationship is not significant, however, we do see a pattern over a 3 year period that Brewer sparrow adult density is 1.5-2 times higher than nest density. We made changes to the text, especially our interpretation in the Discussion.

Line 396: I think you could strengthen your manuscript by more (and earlier) discussion about why we care about density dependence. You seem to be arguing from a management perspective, but the links haven’t been fully developed or explained. Some additional explanation would help pull that together and help the reader understand why we should care about this paper.

We concur with the reviewer and made changes to reflect why we care about density dependence.

Reviewer #2: Comments to Authors, PONE-D-22-21628

General Remarks:

Note: I also reviewed a previous submission of this manuscript for a different journal. The previous and current versions of the manuscript are very similar, as are my comments and concerns. I admittedly found it disconcerting that the authors apparently elected not to incorporate most suggestions or address the concerns articulated in the previous round of reviews, as peer reviews necessitate considerable effort and time on the part of reviewers. Despite the similarities between versions, however, I re-read and evaluated the current version thoroughly.

The authors of this paper assessed relationships between livestock grazing regimes and songbird demography within a habitat type that has been altered extensively range-wide. Moreover, they attempted to assess relationships between adult density, nest density, and nest survival. I concur that understanding of grazing effects on songbirds and the extent to which density-dependence operates on nest survival (and within which contexts) is lacking. Moreover, understanding the relationship between the number of adults within an area and actual reproductive effort (herein, nest density) is important for assessing how meaningful density is as a proxy within different systems. As such, the authors have identified some important gaps in understanding, both broadly across species, and within the focal system from a conservation perspective.

I also found the writing to be generally clear and concise.

Whereas the spatial replication (n = 80, 25-ha study plots) and study duration (n = 3 years) were impressive, I still have some strong reservations about whether the study approach can suitably address the objectives related to density, and whether the system was a logical one within which to expect and evaluate density-dependence. The study took place within an area that had very low densities of the two focal species compared with other parts of their range. I also think that the authors put forth some explanations (e.g., pairs skipping breeding during some years) that are unwarranted given the approach and data, and could be unnecessarily misleading to readers in terms of furthering understanding of the species. I elaborate on these concerns, and provide some additional suggestions, below.

We appreciate the comments of the reviewer and their assistance in improving our manuscript. We apologize if the reviewer feels we have neglected and not respected their time. However, we believe we have philosophical differences about our paper and what our study can provide in advancing our understanding of songbird populations. We hope the editor considers this difference in your assessment of our resubmission.

Specific Comments:

With respect to the tests of grazing effects, I did not find any description of the extent of grazing pressure (e.g., average number of cattle per ha or similar). Without such information, readers cannot compare grazing levels across spatiotemporal contexts. Even providing a range of typical livestock densities present in the study plots would be helpful. Without this information, readers cannot assess whether the grazing regimes were comparatively low, moderate or high. Perhaps grazing was not influential simply because livestock densities were low enough to have minimal influence on habitat metrics that could influence food or nest predation rates for breeding songbirds, whereas at higher grazing levels negative effects could result.

We agree with the author that to address grazing, we need some grazing metrics. We made substantial changes, given we did not have any grazing metrics to make this evaluation.

Indeed, the introduction emphasizes habitat changes in the sagebrush steppe, and the potential influence of grazing regimes, whereas the grazing aspect seems sort of “tacked on” when it comes to the articulation of the study questions at the end of the introduction.

We agree to this comment which is similar to reviewer 1 and have re-worked the manuscript, particularly the Introduction and Discussion.

I also found myself craving more information about the habitat structure/composition of the SGI versus non-SGI plots. Being able to more explicitly link the two grazing regimes to potential differences in resultant habitat characteristics and then those to avian responses would be more compelling than simply comparing the effects of the two grazing treatments (especially without the inclusion of grazing intensity). This would be especially helpful for understanding why “grazing regime had a negative effect on adult pair density for Brewer’s sparrows” (lines 289 – 290).

We agree and have made changes throughout the manuscript.

I think more ecological context would strengthen the premise of the study. For example, the primary source of nesting mortality was nest predation. How/why did you anticipate that grazing in your system could affect egg and/or nestling predation rates (lines 95 – 96)? Do you have a sense for the primary nest predator species?

Our paper is not about nest predation rates or cause-specific mortality of nests. Thus we do not believe adding this information is relevant to our paper. Plus, we did not specific identify nest predators in our study area, which may be several different mammals (e.g., coyotes, skunks, foxes), birds (e.g., ravens, cowbirds), and reptiles (numerous predatory snakes in the study area).

The sample sizes of nests (N = 59 BRSP and 65 VESP) located via transect surveys were extremely low relative to other studies, especially considering the timespan (3 years) and n = 80 plots. This further exemplifies my confusion as to why the authors envisioned that their system was a logical one in which to expect density-dependent demographic responses. Certainly, local habitat quality influences the potential of an area to support individuals and the point at which density-dependence would manifest, but the authors do not describe key attributes such as mean/variation of shrub cover and height for comparison, other than to say in the Study area section that the area consists of “less shrub and forb cover and shorter shrubs than other sagebrush steppe areas”. I think at a minimum, in the discussion the authors should address their densities/nest sample sizes compared with other studies that have monitored BRSP and VESP nests, and more thoroughly address the types of systems in which density-dependent nest survival is most likely to operate.

While we do not disagree with the reviewer's comment that our study system may not have been the most ideal, we are not sure why we should state that other systems would be more suited to address our question about density-dependent. We did the best we could, given our system. And the reality is that there have been many studies on density-dependence, especially on ungulates, that still end up stating their study system may be lacking information to look at density-dependence. Density-dependence is complex, and more samples/different densities do not make it clearer to address. Our study points out that we need to consider this form of population regulation in songbirds, a point that has seldom been stated in songbirds. 

Nest survival analysis section: I do not fully understand from the explanation provided (or supplementary data) how the two nest density covariates were calculated, with which to assess the effect of nest density on nest survival. Please clarify. Were the covariates that you used the distance to the nearest nests of each species? If so, then would this not more likely be reflective of spatial autocorrelation in rates of nest predation rather than nest densities per se?

We did not look at distance to nearest nests in this manuscript. Thus, we are unsure of the point of this comment.

I did not find figure 1 to be particularly necessary, as it does not convey useful spatial information beyond that described in the text.

We concur and removed figure 1.

I suggest standardizing your density data in figures to per square ha to be more directly comparable to other studies.

We do not disagree with the reviewer that changing the units would be best for comparison. But the focus of our paper is not to compare it to other studies. Our focus is to explore density-dependence regulation in songbirds.

Because ratios of the densities of adults to nests (especially for BRSP) were skewed towards adults, the authors concluded (1) that some birds may forgo breeding within some years, and (2) evidence of density-dependence. I have many reservations about these conclusions, as fleshed out in the following bullets:

• There are many reasons in my estimation for why the number of nests observed in an area should be lower than the number of adults present, as not every pair has an active nest at a given point in time. Indeed, in this study, approximately half of nests failed, often during the incubation period, meaning that at any given time, some parents within an area would be between nesting attempts. Both species also are multi-brooded, so some parents would have been between nesting attempts even after the first brood fledged.

We do not disagree and state a few times that multi-brooding may be occurring and why we see the pattern we do.

• Second, early-stage (build, pre-lay and lay) nests, during which adults often are not present, likely would not have been detected using the nest-searching method employed, which would bias nest numbers low. Both species will re-nest multiple times following nest failures, so early-stage nests exist throughout much of the season.

We do not disagree with the author, but we define a nest differently. We only considered a nest as a structure with at least one egg. Thus, pre-nest (based on our definition of nest) breeding activity was not part of our study. We further believe what the reviewer refers to is another component of the reproductive effort that should be considered separately.

• The assumption of a 1:1 sex ratio for observed adults also is potentially problematic, because detection probabilities of males (who sing) versus females (more cryptic) can be very different which could bias the number of actual pairs high.

We do not disagree that a 1:1 sex ratio is likely not correct. It is not correct for most species. However, it’s the best we could do based on the information available. We could not locate any study that suggested a sex ratio. Further, we believe that males sign and females listen (or sing less) is not proven but highly suggested for these two songbird species. 

• The timing of relating the number of adults to nests within an area also matters. Whereas the authors indicated that surveys for adults and nest-searching were somewhat temporally matched, it was unclear at the analytical stage whether the authors compared the density of birds observed with nest densities within the same time period (e.g., week) or averaged across the season. Pairs may not have had an active nest at any given time for reasons described above. Summarizing the data across the entire season could therefore introduce considerable noise into the analyses.

We do not disagree with the reviewer and we believe the ‘noise’ they state is an aspect the variability we estimate in our estimates of nest density and adult density. 

• The concept that BRSPs would skip breeding within a year does not make sense for a relatively short-lived bird. Certainly, some males pair later than others, and some may not pair at all, especially if they are not able to procure and defend a suitable territory. Without monitoring individually-marked birds, however, one cannot discern whether particular males or females paired or initiated a nest at any point within a season or whether individuals whose first nests failed tended to disperse within a season to try to breed elsewhere. The authors cite one paper from forty years ago to support the idea of a large proportion of male BRSP not initiating nests within a season, but those authors similarly did not mark individual birds. Suggesting that the focal species “may not breed” every year as a tactic could therefore be very misleading.

In general, ecological theory states that long-lived species should be selective about when they breed, and short-lived species should reproduce as often as possible. It has been stated, but not quantified, that Brewer’s sparrows and vesper sparrows are short-lived (for instance, the reviewer states BRSP are short-lived). However, as the reviewer states, without individually-marking birds and following them, we do not really know if they are long-lived or short-lived. Its been assumed that they are short-lived based on their size, but there is no clear evidence. As information because available, many small birds, including songbirds, are known (there is clear evidence) to live for approximately a decade (e.g., Eastern Bluebirds, American Robins, Yellow Warblers) which is longer than that ecological theory would suggest.

---

## [Decision Letter · Decision Letter 1]

2 Jan 2023

PONE-D-22-21628R1Density dependence of songbird demographics in grazed sagebrush steppePLOS ONE

Dear Dr. Ruth,

Thank you for submitting your manuscript to PLOS ONE. After careful consideration, we feel that it has merit but does not fully meet PLOS ONE’s publication criteria as it currently stands. Therefore, we invite you to submit a revised version of the manuscript that addresses the points raised during the review process.

I appreciate the authors’ attention in addressing many of the comments and suggestions by the original reviewers.  This paper could provide a welcome addition to our understanding of the demography on songbirds but not in its current form.   The revisions have improved the clarity to focus on density dependence and strengthening the main conclusions of the paper.  There are, however, a number of issues that have not been addressed that undermine the presentation of the work.  The revision does a good job of refocusing the paper on demography based on SGI status, which is a useful only, though, if there is a clear depiction of the characteristics associated with SGI versus non-SGI plots as Reviewer 2 points out.  In addition, there are a number of places where the language suggests inferences that do not seem to be supported by the data being presented and seems like unwarranted extrapolation.  Reviewer 2 highlights these in detail and they need to be addressed to prevent undermining any of your defendable conclusions.  These unsupported extrapolations are the reason a rejection is suggested so they are critical to address.  You need to make sure that you are not overstating what are merely hypotheses as to why your results may differ from other studies and to think about the implications of these if they are taken as factual rather than the speculation they are, particularly by practitioners.  Also, please make sure you have been thorough in including all relevant literature, as this will not only provide additional support for your findings but place it within the context of what is known with confidence. 

I encourage you to go back to the original reviewer comments to see what might have been missed.  This is especially important as Reviewer 3, who provides a fresh perspective, noted that there are still a number of unaddressed issues from the original critique.  They make a number of very useful and substantive suggestions on how to address a number of them.  I agree with Reviewer 3’s suggestion that a supplemental table with clarification of the variables that would be very helpful and they also point out a few other ways to present some of your findings that may be more impactful.  Both reviewers make detailed suggestions and indicate areas that are unclear or misleading that must be addressed.  They have been very thorough and give concrete suggestions on improving language highlighting areas where questions remain.  There are still a number of clarifications needed in the methodology and the interpretation of the results before this manuscript is suitable for publication.  While both reviewers and I feel there is value in this research, it is not well defended in its current version and needs substantial revision.

We look forward to receiving your revised manuscript.

Kind regards,

Karen Root, Ph.D.

Academic Editor

PLOS ONE

Reviewers' comments:

Reviewer's Responses to Questions

**Comments to the Author**

1. If the authors have adequately addressed your comments raised in a previous round of review and you feel that this manuscript is now acceptable for publication, you may indicate that here to bypass the “Comments to the Author” section, enter your conflict of interest statement in the “Confidential to Editor” section, and submit your "Accept" recommendation.

Reviewer #2: (No Response)

Reviewer #3: (No Response)

2. Is the manuscript technically sound, and do the data support the conclusions?

Reviewer #2: No

Reviewer #3: Partly

3. Has the statistical analysis been performed appropriately and rigorously? 

Reviewer #2: Yes

Reviewer #3: No

4. Have the authors made all data underlying the findings in their manuscript fully available?

Reviewer #2: Yes

Reviewer #3: Yes

5. Is the manuscript presented in an intelligible fashion and written in standard English?

Reviewer #2: Yes

Reviewer #3: Yes

6. Review Comments to the Author

Reviewer #2: The authors have softened the focus on grazing effects throughout the paper which was an improvement and is logical given that information on grazing intensity was not available.

The new focus on the SGI, however, seems a bit undeveloped. For example, the authors state in the introduction (lines 85 – 86) that: “However, this conservation program may influence density dependent responses in sagebrush steppe songbirds.” How? Why? What specifically about the implementation of the SGI and how it tends to affect sagebrush habitats make it potentially influential on density dependence per se?

Moreover, without a better understanding of how SGI versus non-SGI study plots differed in ways pertinent to nesting songbirds, such as resultant habitat structure or composition, I am not convinced that the comparison is particularly useful for practitioners.

The authors also state in the abstract that they “found evidence for density dependence when assessing the ratio of adult pair density to nest density.” How? In what way? Demonstrating that adult birds were detected more frequently than nests within a particular time period does not necessarily suggest that density-dependence was operating. Moreover, birds were not individually marked so how were pairs definitively delineated?

First paragraph in the introduction: Logical flow could be improved, as the scope of inference jumps around a bit. For example, the authors begin with sagebrush songbirds and then zoom back out to discuss density dependence in general. I suggest beginning with the bigger-picture context then leading up to the specific system and questions.

The following sentence in the introduction: “Density dependence has a paucity of information regarding how it affects vital rates in avian populations” does not make sense as written.

Lines 91 – 92: What was the biological rationale for your a priori hypothesis that nest survival for sagebrush songbirds is density-dependent? I still don’t understand why with such low densities of a species one would expect density-dependent responses. Two BRSP nests within a 25-ha plot, for example, is very sparse.

I still do not agree with the assertion, moreover, that two adult birds should necessarily equate to one nest within a set timeframe an, for many reasons that I articulated at length in my previous reviews. This does not negate, however, the potential importance of evaluating how the density of adults (as typically assayed by monitoring programs etc.) generally relates to nesting densities within an area.

Still unclear to me is the temporal period across which nest densities were compared with adult densities. Were the data summarized within a plot and year? If so, did you relate the average or maximum number of adults and nests detected within a plot and year to one another?

Do you think the detection probabilities of nests (0.38) and adults (0.48) was sufficient (and sufficiently similar) to justify comparison?

Line 228: what is meant by “nesting territories?”

Lines 306 – 314: I think referring to density-dependent population regulation is outside the scope of the study. Many factors can influence population growth, and some vital rates have much higher sensitivity to actual lambda than others.

Lines 307 to 314: The first part of this text is written in the form of predictions which better belongs at the end of the introduction when questions/hypotheses/predictions are presented. The last sentence is interpretation which typically belongs in the discussion section.

Discussion, first sentence: Again, why (in this system) did you hypothesize that nest density would be density-dependent? Nest densities were very low, both compared to other sagebrush steppe and other habitat types. Without more biological explanation the focus is not convincing.

Lines 327 – 328: Just because nest densities were positively correlated with nest survival does not mean it had anything to do with density, per se. Alternatively, more birds may have been attracted to settle in areas of higher habitat quality.

Lines 336 – 338: I remain steadfast in my opposition to the conclusion that just because more adult birds were detected than nests suggests that some birds may not have initiated a nest within a given season. There just simply is not enough evidence to conclude this with the relatively low detection probabilities of nests and adults, and the inability to differentiate among individuals, in addition to all of the other arguments I suggested in my last reviews of this paper. This also is not a necessary conclusion to include in terms of the primary objectives.

Lines 383 to 383: Most arid shrubland and grassland songbird nestlings do not elicit begging calls, which parallels their general propensity towards crypticity.

Reviewer #3: General comments:

Although I did not review the original submission, from the track changes and response to reviewer comments, I found the manuscript revision to be greatly improved from the original submission. However, I do not feel that the authors fully addressed all of the original reviewer comments and I would like to see those better addressed prior to publication. I do feel that the authors provided the caveats appropriately that the original Reviewer 2 requested, although they could be wordsmithed for clarity. I have listed the specific original comments still needing to be addressed below. These are followed by specific comments from myself. I found the article on the whole to be well written and concise, but I have three major concerns. The first is the use of apparent nest success. I agree with the original Reviewer 1 that it is not useful, and I would go further and caution that it could be harmful/misleading. The authors' response that their sample size was too low to use other methods was unsatisfactory to me. I would like the authors to use a more reliable estimator of nest success. It does not have to be logistic exposure, but it should not be apparent nest success. My second concern is that the results and discussion for the nest survival section do not match the table S3. I am unsure if this is a typo in the table or if something else is going on, but as presented the results and discussion are not supported by the modelling results. This is specific for Vesper's sparrow where the authors say the top model includes nest densities from both species but list the top model as Stage + Year + SGI with no mention of density. Lastly, I have some concerns regarding their methods that need clarifying, including how accurate their distances were that were included in the distance sampling, and the appropriateness of excluding auditory detections from their surveys, which is in contradiction to the papers they cite. I do believe that their final conclusion that adult pair densities are not a good proxy for species such as Brewer's sparrows, has management implications and would contribute greatly to the literature.

Original reviewer comments not fully addressed:

Line 137: What other biotic factors did you examine? Reading further, it isn’t clear that you

incorporated any additional variables beyond grazing regime, year, and date. The lack of any

habitat variables seems like a problem, particularly for nest survival.

We recognize that additional biotic factors that were not collected may be influencing nest

survival.

This was not addressed in the revised manuscript. The authors state the tested "biotic factors" but did not list them I the text.

Furthermore, Reviewer 2's comments below were not fully addressed:

"Nest survival analysis section: I do not fully understand from the explanation provided (or

supplementary data) how the two nest density covariates were calculated, with which to assess

the effect of nest density on nest survival. Please clarify. Were the covariates that you used the

distance to the nearest nests of each species? If so, then would this not more likely be reflective

of spatial autocorrelation in rates of nest predation rather than nest densities per se?

We did not look at distance to nearest nests in this manuscript. Thus, we are unsure of the point

of this comment."

and

"The sample sizes of nests (N = 59 BRSP and 65 VESP) located via transect surveys were

extremely low relative to other studies, especially considering the timespan (3 years) and n = 80

plots. This further exemplifies my confusion as to why the authors envisioned that their system

was a logical one in which to expect density-dependent demographic responses. Certainly, local

habitat quality influences the potential of an area to support individuals and the point at which

density-dependence would manifest, but the authors do not describe key attributes such as

mean/variation of shrub cover and height for comparison, other than to say in the Study area

section that the area consists of “less shrub and forb cover and shorter shrubs than other

sagebrush steppe areas”. I think at a minimum, in the discussion the authors should address their

densities/nest sample sizes compared with other studies that have monitored BRSP and VESP

nests, and more thoroughly address the types of systems in which density-dependent nest

survival is most likely to operate."

I would like a table in the supplementary material listing variables they tested and their definitions or more clarity as they were calculated. I concur with reviewer 2 that there is a lack of explanation as to how the nest density covariates were calculated. The authors only state in a footnote "

3Interspecific Density = combined nest density of Brewer’s and vesper sparrow nests 4Intraspecific Density = nest density of respective species" I would like to see a better explanation of the calculations and how this was included in the analysis for the density dependence.

Line 307: reviewer 1 questioned the use of apparent nest survival. I concur with the problem (also see my comment for lines 261-264 in the new manuscript.) The author's response that the sample size was too low is not a good enough reason to include apparent nest success in a manuscript when we know how flawed it is. Unsuccessful nests are less likely to be found than successful. I would like the authors to at least use the Mayfield estimator or a different method if the logistic exposure method does not have the sample size at the plot level.

Specific comments:

Line 110: authors switch from past tense to present tense

Line 113: duplicate comma

Line 118: missing the word "Were" …. "Of the 80 plots, 40 [were] privately managed…"

Lines 126-127: Authors state they examined other "biotic and abiotic factors…" please list them, or reference a table. I only see stage, year, SG, day of year and density referenced in the appendix. What about vegetation variables? It appears the authors did not collect vegetation data, but GIS level data would give some ideas of differences across sites, especially for sagebrush cover. I suspect that differences in Brewer's sparrow numbers have more to do with the shrub cover than they do with density dependence. In lines 111-112 they mention there is less shrub cover, but I would like to see some sort of summary. As a sagebrush-obligate species, not including any measure of sagebrush cover seems very problematic when assessing density dependence, as the number of nests/individuals that the area could support is likely related to the shrub and forb cover among other factors.

Line 131: how big are the "plots?" 500 m long at least I assume… but there is no mention of how many transects were placed parallel 100 m across, or how wide the plots are. Would be helpful to get a feel for how well the plots were surveyed, as the nest densities are rather low.

Line 140: The authors indicate they used GPS coordinates to measure the distance between the transect line and nests. How far on average was this distance and what is the accuracy of the GPS used? If observers were walking 5m from the transect line, I imagine that might be the average, and that coincidentally is the accuracy of many GPS units. Why did the observers not carry measuring tapes and measure the distance? If the distances from the transect were typically much greater than the accuracy of the GPS then I can buy this method, but for a distance sampling analysis I find it concerning if I cannot trust the accuracy of the distances measured.

Line 141: What was the range of dates these surveys started? A Day of year range- which the authors call Julian date would be helpful if surveys changed based on year.

Line 147: abandonment usually occurs earlier than brood rearing unless the adult is killed, how did you determine abandonment during the incubation stage? Time to estimated hatch? Several visits without observing an adult?

Lines 156-157: immediately before nest surveys: this read to me as the same day, but the next sentence implies it was the next, consecutive day, and the latter half of line 157 has a range of 1-3 days. These sentences are confusing. Maybe list how often percentage-wise they were conducted "immediately before" and better define what that means. Same day? Different days?

Line 159: The authors indicate they use the double observer method developed by Nicholas et al but indicate they only recorded visual detections (line 168) of the songbird species studied. This is not actually the method described by Nicholas et al, as in their methods they identify all seen and heard: "The primary observer identifies all birds seen and heard and communicates (via speech and gesture) to the secondary observer the species detected and the direction and general distance of the detection." Why did the authors only use visual identification of species when their cited source uses auditory? This strikes me as potentially problematic considering the cryptic nature of these songbirds. This difference in method may have affected the density calculations used in density dependence variables. The citation #23 also included auditory detections in their analyses.

Lines 160-161: authors reference 2 papers by the same author in the same system, but one is an occupancy paper, which is a different analysis than the authors here use, the second did indicate they included auditory detections (see my comment above). Furthermore, these papers are for burrowing owls and plovers, which I assume have a rather different type of detection. Burrowing owls are less vocal than sagebrush obligate songbirds and more easily spotted visually. Has this method ever been used for songbirds?

Line 163: this citation is not correct for this statement as it is not the original and refers to "(Scott and others 1981)" It is also discussing the 125m distance in a different context- establishing the minimum distance between surveys. The cited reference recommends a larger distance in open landscapes. It seems the authors may have double counted adults per language from their cited reference: "In addition, the maximum detection of virtually all individuals of most species is less than 250 m (Wolf and others, in this volume). In open environments, this minimum distance should be increased due to the greater detectability of birds. " I would like the authors to better explain why that distance was chosen- do they think they double counted birds? Why the U shape instead of a straight line that would prevent duplication?

Lines 178-179: why were these the candidate models included? Citation 28 is a nest survival analysis and does not make sense to cite here.

Lines 180-181: I don't see a table 1, which is referenced here in the text

193-195: worded a bit confusingly- was year a random effect or just plot?

Lines 196-198: I would like more details on how the nest density interspecific and intraspecific variables were calculated. Were these per site? Authors mention per year, but summarized across what scale? I need more information.

Lines 200-202: were there other species' nests present that could be contributing to interspecific density dependence? Sage thrasher, sagebrush sparrow, etc? Do the authors feel this value fully encompassed interspecific density dependence?

Line 204: interactions implied multiplicative interactions to me, can you specify if the only interactions tested were additive combinations? Or did you test multiplicative and not show them in the table?

Line 210: S3 table is for nest survival- what is the author trying to reference here? Is there a missing table for adult density?

Lines 227-228: given these violations of that assumption, do the authors still feel the results are valid or useful? During the adult surveys was the behavior of the bird noted? If so this might give some indication of sex (perched and singing versus silent or only producing call notes). As written the reader is left wondering if the results are useful. Perhaps language as to how these violations would effect results (artificially increase or decrease) could be helpful.

Lines 248-249: lack of a difference does not mean Vesper's sparrow failed to respond to the SGI program, but it could be due to low sample size. 65 nests across 80 plots over 3 years is relatively low nest density.

Line 251: perhaps conservation regime instead of grazing regime? To be more consistent with your manuscript? It would also be really helpful if you indicated if the CIs do or don't overlap somehow, like with an * Just curious- but why did you summarize across the SGI and non-SGI but not across years? Seems like that would be the more interesting comparison (SGI vs non-SGI summarized across 2016-2018)

Line 260: oh wow, your nest numbers are not as low as I thought- seems like you found a lot of nests opportunistically. Why do you think that is? Were these missed in your plots during the nest searching or were they outside of the search area? Over half of your Vesper nests were found opportunistically, that is surprising to me. This might be helpful to mention with a (n= XX) in the results following your words opportunistically and distance sampling.

Lines 261-264: seems like you shouldn't use apparent nest success here. Why not use your modelled data to provide an estimate of overall nest success or at least Mayfield's. Apparent nest success is often more misleading than helpful. I have never seen nest survival lumped across species, especially ones quite different. This seems odd for lines 263 and 264. Predation rates would be more informative by species, not lumped together.

Lines 270-271: this statement is not supported by your AIC table. Table S3 does not indicate that density is in the top model. You also mention total nest density here and define it as nest density for both species. You define interspecific density that way in your table, but that is shown in the second model, not the top model.

Line 285: ah, this is where you say the size of the plot. Please include this in the methods section.

Line 307: See above comment- I agree with reviewer #1 and do not feel this was addressed in this revision

Line 326: Perhaps there is an error in the S3 table? As this statement is not supported by your modelling results where the top model is listed as Stage + Year + SGI. The top model is also competitive with two others, which deserves discussion. I would need to see a supplementary table with the parameter estimates of the top model (or models if several are within 2AIC) to be sure of the results and interpretation the authors are making

Lines 328-329: inverse density dependence is an odd way of putting this. It appears that many individuals are selecting areas of high quality habitat that are able to support more nests, and those who chose these areas are rewarded with higher success rates. I assume if the authors had any type of habitat quality data they could dig into what is going on here to promote higher nest survival in such areas. To describe it as inverse density dependence suggests the Allee effect where populations at low numbers increase in growth rate/ nest survival rate- which is not the case here. I suggest re-wording this sentence.

Lines 330-331: there needs to be a bit more to finish this thought- see reviewer #1 comment from lines 353-355. I agree with the reviewer that you seem to be implying "that adults may be less vigilant if other birds

are around, by which I think you’re implying that they could be more successful because they don’t have to use as much of their energy reserves being vigilant?" I think you only addressed the second portion of their original comment and changed the next sentence to say additionally instead of in this way, but you didn't finish your thought.

Lines 333-335: see previous comment about your interspecific density metric

Lines 344-349: I see this was added to address Reviewer 2's comments. I think this does that, but could be re-worded for clarity using what the hypothesis is rather than referring to rejecting your hypothesis and having the reader hunt back through the text for what the hypothesis was exactly.

Lines 350-353: I think I understand what you mean, but consider re-wording for clarity, as to what the considerations should be- just that managers know there are birds that are non-breeders or migrants? And not assume the population is doing as well? What's the implication?

Lines 358-359: consider re-wording "The exact mechanisms that caused the differences for Brewer's sparrows" reads odd. Which differences? In nest density?

Line 360-362: perhaps you are missing a word here? I think you are trying to say that Nest density and nest success did not differ for the thick billed longspur

Line 362: similarly

Lines 360-363: Maybe a sentence or a few words connecting these thoughts that other species in the area were not affected by SGI status.

Lines 395-397: I'm not sure you need this statement. Now that the paper is less focused on grazing it seems a bit out of context.

7. PLOS authors have the option to publish the peer review history of their article (what does this mean?). If published, this will include your full peer review and any attached files.

Reviewer #2: No

Reviewer #3: No

---

## [Author Response · Author response to Decision Letter 1]

19 Jul 2023

Reviewer #2: The authors have softened the focus on grazing effects throughout the paper which was an improvement and is logical given that information on grazing intensity was not available.

The new focus on the SGI, however, seems a bit undeveloped. For example, the authors state in the introduction (lines 85 – 86) that: “However, this conservation program may influence density dependent responses in sagebrush steppe songbirds.” How? Why? What specifically about the implementation of the SGI and how it tends to affect sagebrush habitats make it potentially influential on density dependence per se?

We adjusted the phrasing and re-worded our reasons for testing SGI implementation. 

Moreover, without a better understanding of how SGI versus non-SGI study plots differed in ways pertinent to nesting songbirds, such as resultant habitat structure or composition, I am not convinced that the comparison is particularly useful for practitioners.

We included additional vegetation metrics to quantify SGI vs non-SGI study plots and included these covariates in our analyses when relevant. 

The authors also state in the abstract that they “found evidence for density dependence when assessing the ratio of adult pair density to nest density.” How? In what way? Demonstrating that adult birds were detected more frequently than nests within a particular time period does not necessarily suggest that density-dependence was operating. Moreover, birds were not individually marked so how were pairs definitively delineated?

We concur with the reviewer and have rewritten the abstract.

First paragraph in the introduction: Logical flow could be improved, as the scope of inference jumps around a bit. For example, the authors begin with sagebrush songbirds and then zoom back out to discuss density dependence in general. I suggest beginning with the bigger-picture context then leading up to the specific system and questions.

The first paragraph has been rewritten for improved logical flow. 

The following sentence in the introduction: “Density dependence has a paucity of information regarding how it affects vital rates in avian populations” does not make sense as written.

This sentence has been reworded. 

Lines 91 – 92: What was the biological rationale for your a priori hypothesis that nest survival for sagebrush songbirds is density-dependent? I still don’t understand why with such low densities of a species one would expect density-dependent responses. Two BRSP nests within a 25-ha plot, for example, is very sparse.

We have added text to highlight the lack of understanding of density dependence in any population size and density level. Additionally, an alternative hypothesis of no relationship has been added. 

I still do not agree with the assertion, moreover, that two adult birds should necessarily equate to one nest within a set timeframe an, for many reasons that I articulated at length in my previous reviews. This does not negate, however, the potential importance of evaluating how the density of adults (as typically assayed by monitoring programs etc.) generally relates to nesting densities within an area.

While we agree that we may have sometimes counted two birds of the same sex as a nesting pair, we do not agree that, for instance, we should always count a singing bird as a male and a silent bird as a female. We have noted instances of female birds singing songs in other studies we have conducted with banded birds. Moreover, how do we then count birds giving chip calls? No matter how we decide to delineate nesting pairs, we would always be making some assumption about the sex of the bird based on potentially spurious cues. Therefore, we opted for the option that was most straightforward to implement across all surveys.

Still unclear to me is the temporal period across which nest densities were compared with adult densities. Were the data summarized within a plot and year? If so, did you relate the average or maximum number of adults and nests detected within a plot and year to one another?

In the methods section we added text for clarity as to how nest and adult densities were calculated. They were calculated at the plot level for each year. Those values were then used to look at the relationship between nest density and adult density at the year-specific plot level. 

Do you think the detection probabilities of nests (0.38) and adults (0.48) was sufficient (and sufficiently similar) to justify comparison?

We do feel the estimates of detection were sufficient to make the comparison. Detection is incorporated into the estimates of nest density and adult density, and their corresponding levels of precision. Thus, we believe our estimates of nest density and adult density are more accurate than not accounting for detection. 

Line 228: what is meant by “nesting territories?”

This has been reworded for clarity. 

Lines 306 – 314: I think referring to density-dependent population regulation is outside the scope of the study. Many factors can influence population growth, and some vital rates have much higher sensitivity to actual lambda than others.

Due to the new analysis results, this paragraph was removed. 

Lines 307 to 314: The first part of this text is written in the form of predictions which better belongs at the end of the introduction when questions/hypotheses/predictions are presented. The last sentence is interpretation which typically belongs in the discussion section.

Due to the new analysis results, this paragraph was removed. 

Discussion, first sentence: Again, why (in this system) did you hypothesize that nest density would be density-dependent? Nest densities were very low, both compared to other sagebrush steppe and other habitat types. Without more biological explanation the focus is not convincing.

We added context and biological explanation. 

Lines 327 – 328: Just because nest densities were positively correlated with nest survival does not mean it had anything to do with density, per se. Alternatively, more birds may have been attracted to settle in areas of higher habitat quality.

We are unclear by the reviewer’s point, but we will try to address. In the manuscript, we state “Higher quality habitat sites are associated with higher vital rates, including survival and reproduction (Rodenhouse et al. 1997)”. Thus, if more birds are attracted to higher habitat quality for nesting, then more nests and higher nest success. Its when high quality sites are filled, and individuals forgo nesting or use suboptimal habitat, nest density and nest success will decrease.

Lines 336 – 338: I remain steadfast in my opposition to the conclusion that just because more adult birds were detected than nests suggests that some birds may not have initiated a nest within a given season. There just simply is not enough evidence to conclude this with the relatively low detection probabilities of nests and adults, and the inability to differentiate among individuals, in addition to all of the other arguments I suggested in my last reviews of this paper. This also is not a necessary conclusion to include in terms of the primary objectives.

We acknowledge some of the reviewer’s concerns. We re-organized and re-worded our text to soften the language we had earlier. However, we are not the only study suggesting that not every Brewer’s sparrow breeds every year. We include a reference (Reynolds 1981) in the manuscript. Further, we are unclear on the logic of low detection probabilities as a reason there is not some evidence. We estimated detection (and its precision) and accounted for detection in our estimates of nest density and adult density and their corresponding confidence intervals. Our detection probabilities could have been lower than they were, but we could still have the same level of precision in our estimates. Similarly, detection could have been rather high (e.g., 0.90), and we still could have the same level of precision in our estimates suggesting that adult density was higher than nest density. Thus, we are unclear of the reviewer's point of ‘relatively low detection probabilities’ is a reason we can NOT suggest there are more adults than nests for Brewer’s sparrows in our study area.

Lines 383 to 383: Most arid shrubland and grassland songbird nestlings do not elicit begging calls, which parallels their general propensity towards crypticity.

This section of the discussion was removed. However, observers in the field occasionally heard begging calls when approaching nests monitored in this study. 

Reviewer #3: General comments:

Although I did not review the original submission, from the track changes and response to reviewer comments, I found the manuscript revision to be greatly improved from the original submission. However, I do not feel that the authors fully addressed all of the original reviewer comments and I would like to see those better addressed prior to publication. I do feel that the authors provided the caveats appropriately that the original Reviewer 2 requested, although they could be wordsmithed for clarity. I have listed the specific original comments still needing to be addressed below. These are followed by specific comments from myself. I found the article on the whole to be well written and concise, but I have three major concerns. The first is the use of apparent nest success. I agree with the original Reviewer 1 that it is not useful, and I would go further and caution that it could be harmful/misleading. The authors' response that their sample size was too low to use other methods was unsatisfactory to me. I would like the authors to use a more reliable estimator of nest success. It does not have to be logistic exposure, but it should not be apparent nest success. My second concern is that the results and discussion for the nest survival section do not match the table S3. I am unsure if this is a typo in the table or if something else is going on, but as presented the results and discussion are not supported by the modelling results. This is specific for Vesper's sparrow where the authors say the top model includes nest densities from both species but list the top model as Stage + Year + SGI with no mention of density. Lastly, I have some concerns regarding their methods that need clarifying, including how accurate their distances were that were included in the distance sampling, and the appropriateness of excluding auditory detections from their surveys, which is in contradiction to the papers they cite. I do believe that their final conclusion that adult pair densities are not a good proxy for species such as Brewer's sparrows, has management implications and would contribute greatly to the literature.

Original reviewer comments not fully addressed:

Line 137: What other biotic factors did you examine? Reading further, it isn’t clear that you incorporated any additional variables beyond grazing regime, year, and date. The lack of any habitat variables seems like a problem, particularly for nest survival. We recognize that additional biotic factors that were not collected may be influencing nest survival.

This was not addressed in the revised manuscript. The authors state the tested "biotic factors" but did not list them I the text.

Furthermore, Reviewer 2's comments below were not fully addressed:

We have included further biotic and abiotic factors as well as tables in the appendix to explain those covariates. 

"Nest survival analysis section: I do not fully understand from the explanation provided (or supplementary data) how the two nest density covariates were calculated, with which to assess the effect of nest density on nest survival. Please clarify. Were the covariates that you used the distance to the nearest nests of each species? If so, then would this not more likely be reflective of spatial autocorrelation in rates of nest predation rather than nest densities per se?

We did not look at distance to nearest nests in this manuscript. Thus, we are unsure of the point of this comment." and "The sample sizes of nests (N = 59 BRSP and 65 VESP) located via transect surveys were extremely low relative to other studies, especially considering the timespan (3 years) and n = 80 plots. This further exemplifies my confusion as to why the authors envisioned that their system

was a logical one in which to expect density-dependent demographic responses. Certainly, local habitat quality influences the potential of an area to support individuals and the point at which density-dependence would manifest, but the authors do not describe key attributes such as mean/variation of shrub cover and height for comparison, other than to say in the Study area section that the area consists of “less shrub and forb cover and shorter shrubs than other sagebrush steppe areas”. I think at a minimum, in the discussion the authors should address their densities/nest sample sizes compared with other studies that have monitored BRSP and VESP nests, and more thoroughly address the types of systems in which density-dependent nest survival is most likely to operate."

I would like a table in the supplementary material listing variables they tested and their definitions or more clarity as they were calculated. I concur with reviewer 2 that there is a lack of explanation as to how the nest density covariates were calculated. The authors only state in a footnote "

3Interspecific Density = combined nest density of Brewer’s and vesper sparrow nests 4Intraspecific Density = nest density of respective species" I would like to see a better explanation of the calculations and how this was included in the analysis for the density dependence.

This table was edited for clarity and an additional supplemental table was added explaining the variables used. 

Line 307: reviewer 1 questioned the use of apparent nest survival. I concur with the problem (also see my comment for lines 261-264 in the new manuscript.) The author's response that the sample size was too low is not a good enough reason to include apparent nest success in a manuscript when we know how flawed it is. Unsuccessful nests are less likely to be found than successful. I would like the authors to at least use the Mayfield estimator or a different method if the logistic exposure method does not have the sample size at the plot level.

We have removed the apparent nest survival figure and used only estimated nest success estimates. 

Specific comments:

Line 110: authors switch from past tense to present tense

Line 113: duplicate comma

Line 118: missing the word "Were" …. "Of the 80 plots, 40 [were] privately managed…"

These small typos were corrected.

Lines 126-127: Authors state they examined other "biotic and abiotic factors…" please list them, or reference a table. I only see stage, year, SG, day of year and density referenced in the appendix. What about vegetation variables? It appears the authors did not collect vegetation data, but GIS level data would give some ideas of differences across sites, especially for sagebrush cover. I suspect that differences in Brewer's sparrow numbers have more to do with the shrub cover than they do with density dependence. In lines 111-112 they mention there is less shrub cover, but I would like to see some sort of summary. As a sagebrush-obligate species, not including any measure of sagebrush cover seems very problematic when assessing density dependence, as the number of nests/individuals that the area could support is likely related to the shrub and forb cover among other factors.

We included additional biotic and abiotic factors. As the reviewer suggests, we found shrub cover to be predictive on nest success. We also included supplemental tables explaining the covariates used more clearly. 

Line 131: how big are the "plots?" 500 m long at least I assume… but there is no mention of how many transects were placed parallel 100 m across, or how wide the plots are. Would be helpful to get a feel for how well the plots were surveyed, as the nest densities are rather low .

We added text to more clearly define the size of the plots and how the plots were surveyed. 

Line 140: The authors indicate they used GPS coordinates to measure the distance between the transect line and nests. How far on average was this distance and what is the accuracy of the GPS used? If observers were walking 5m from the transect line, I imagine that might be the average, and that coincidentally is the accuracy of many GPS units. Why did the observers not carry measuring tapes and measure the distance? If the distances from the transect were typically much greater than the accuracy of the GPS then I can buy this method, but for a distance sampling analysis I find it concerning if I cannot trust the accuracy of the distances measured.

We added text and references regarding GPS accuracy and the average distance from the transect line that nests were found. 

Line 141: What was the range of dates these surveys started? A Day of year range- which the authors call Julian date would be helpful if surveys changed based on year.

A date range was added in the text. 

Line 147: abandonment usually occurs earlier than brood rearing unless the adult is killed, how did you determine abandonment during the incubation stage? Time to estimated hatch? Several visits without observing an adult?

We added text referring to how we waited until after anticipated hatch date to determine abandonment. 

Lines 156-157: immediately before nest surveys: this read to me as the same day, but the next sentence implies it was the next, consecutive day, and the latter half of line 157 has a range of 1-3 days. These sentences are confusing. Maybe list how often percentage-wise they were conducted "immediately before" and better define what that means. Same day? Different days?

The text has been edited for clarity. 

Line 159: The authors indicate they use the double observer method developed by Nicholas et al but indicate they only recorded visual detections (line 168) of the songbird species studied. This is not actually the method described by Nicholas et al, as in their methods they identify all seen and heard: "The primary observer identifies all birds seen and heard and communicates (via speech and gesture) to the secondary observer the species detected and the direction and general distance of the detection." Why did the authors only use visual identification of species when their cited source uses auditory? This strikes me as potentially problematic considering the cryptic nature of these songbirds. This difference in method may have affected the density calculations used in density dependence variables. The citation #23 also included auditory detections in their analyses.

The citations have been reviewed and fixed. Auditory observations were often used to confirm the identify of the bird visually observed. We also note that Golding and Dreitz 2016 only used visual detections in their analyses. Further, while citation #23 included auditory detections, it was 7% of all observations (119 total observations) which likely would not significantly change that studies results.

Lines 160-161: authors reference 2 papers by the same author in the same system, but one is an occupancy paper, which is a different analysis than the authors here use, the second did indicate they included auditory detections (see my comment above). Furthermore, these papers are for burrowing owls and plovers, which I assume have a rather different type of detection. Burrowing owls are less vocal than sagebrush obligate songbirds and more easily spotted visually. Has this method ever been used for songbirds?

We had reviewed and fixed citations to support our methods. Additionally, a new paper using this method was added.

Line 163: this citation is not correct for this statement as it is not the original and refers to "(Scott and others 1981)" It is also discussing the 125m distance in a different context- establishing the minimum distance between surveys. The cited reference recommends a larger distance in open landscapes. It seems the authors may have double counted adults per language from their cited reference: "In addition, the maximum detection of virtually all individuals of most species is less than 250 m (Wolf and others, in this volume). In open environments, this minimum distance should be increased due to the greater detectability of birds. " I would like the authors to better explain why that distance was chosen- do they think they double counted birds? Why the U shape instead of a straight line that would prevent duplication?

We included additional citations for why this distance was chosen. We train observers on the project to keep track of birds to avoid double counting, including techniques suggested by Nichols et al. (2000).

Lines 178-179: why were these the candidate models included? Citation 28 is a nest survival analysis and does not make sense to cite here.

This was edited for clarity. 

Lines 180-181: I don't see a table 1, which is referenced here in the text

The tables have been reorganized and correctly referenced. 

193-195: worded a bit confusingly- was year a random effect or just plot?

This was edited for clarity. 

Lines 196-198: I would like more details on how the nest density interspecific and intraspecific variables were calculated. Were these per site? Authors mention per year, but summarized across what scale? I need more information.

We have simplified the nest density metric to only be for species-specific nest density and text has been added to explain how those values were used. Nest density was calculated at the year-specific plot level. 

Lines 200-202: were there other species' nests present that could be contributing to interspecific density dependence? Sage thrasher, sagebrush sparrow, etc? Do the authors feel this value fully encompassed interspecific density dependence?

There were very few observations (< 1%) of other sagebrush-steppe songbirds in our study. Our study area is on the sagebrush/grassland fringe in which sagebrush shrubs are most often > 0.5 m. Further, we re-did the analysis and removed text regarding interspecific nest density. 

Line 204: interactions implied multiplicative interactions to me, can you specify if the only interactions tested were additive combinations? Or did you test multiplicative and not show them in the table?

Text and the supplementary table have been edited for clarity. Only additive models were used. 

Line 210: S3 table is for nest survival- what is the author trying to reference here? Is there a missing table for adult density?

The tables have been reorganized and correctly referenced. 

Lines 227-228: given these violations of that assumption, do the authors still feel the results are valid or useful? During the adult surveys was the behavior of the bird noted? If so this might give some indication of sex (perched and singing versus silent or only producing call notes). As written the reader is left wondering if the results are useful. Perhaps language as to how these violations would effect results (artificially increase or decrease) could be helpful.

We have edited the text to be more clear. While we did not note behavioral evidence of individual songbird sex, we do not agree that, for instance, we should always count a singing bird as a male and a silent bird as a female. We have noted instances of female birds singing songs in other studies we have conducted with banded birds. We would always be making some assumption about the sex of the bird based on potentially spurious cues. Therefore, we opted for the option that was most straightforward to implement across all surveys.

Lines 248-249: lack of a difference does not mean Vesper's sparrow failed to respond to the SGI program, but it could be due to low sample size. 65 nests across 80 plots over 3 years is relatively low nest density.

We added text to recognize our small sample size used for nest density estimates.

Line 251: perhaps conservation regime instead of grazing regime? To be more consistent with your manuscript? It would also be really helpful if you indicated if the CIs do or don't overlap somehow, like with an * Just curious- but why did you summarize across the SGI and non-SGI but not across years? Seems like that would be the more interesting comparison (SGI vs non-SGI summarized across 2016-2018)

This table has been fixed using the reviewer’s suggestions.

Line 260: oh wow, your nest numbers are not as low as I thought- seems like you found a lot of nests opportunistically. Why do you think that is? Were these missed in your plots during the nest searching or were they outside of the search area? Over half of your Vesper nests were found opportunistically, that is surprising to me. This might be helpful to mention with a (n= XX) in the results following your words opportunistically and distance sampling.

The nest sample size used in distance sampling could only include nests found on nest searches and only in plots with more than two nests found, which therefore led to more nests that were not used in the distance sampling analysis to be used in the nest survival analysis. Text has been added to clarify this. 

Lines 261-264: seems like you shouldn't use apparent nest success here. Why not use your modelled data to provide an estimate of overall nest success or at least Mayfield's. Apparent nest success is often more misleading than helpful. I have never seen nest survival lumped across species, especially ones quite different. This seems odd for lines 263 and 264. Predation rates would be more informative by species, not lumped together.

The text has been edited for clarity and more specific breakdowns by species. 

Lines 270-271: this statement is not supported by your AIC table. Table S3 does not indicate that density is in the top model. You also mention total nest density here and define it as nest density for both species. You define interspecific density that way in your table, but that is shown in the second model, not the top model.

The tables have been reorganized and correctly referenced. 

Line 285: ah, this is where you say the size of the plot. Please include this in the methods section.

This has been included in the methods section.

Line 307: See above comment- I agree with reviewer #1 and do not feel this was addressed in this revision

This was addressed in this revision.

Line 326: Perhaps there is an error in the S3 table? As this statement is not supported by your modelling results where the top model is listed as Stage + Year + SGI. The top model is also competitive with two others, which deserves discussion. I would need to see a supplementary table with the parameter estimates of the top model (or models if several are within 2AIC) to be sure of the results and interpretation the authors are making

The tables have been reorganized and correctly referenced. 

Lines 328-329: inverse density dependence is an odd way of putting this. It appears that many individuals are selecting areas of high quality habitat that are able to support more nests, and those who chose these areas are rewarded with higher success rates. I assume if the authors had any type of habitat quality data they could dig into what is going on here to promote higher nest survival in such areas. To describe it as inverse density dependence suggests the Allee effect where populations at low numbers increase in growth rate/ nest survival rate- which is not the case here. I suggest re-wording this sentence.

Due to the corrected analysis, we removed text regarding inverse density dependence. 

Lines 330-331: there needs to be a bit more to finish this thought- see reviewer #1 comment from lines 353-355. I agree with the reviewer that you seem to be implying "that adults may be less vigilant if other birds

are around, by which I think you’re implying that they could be more successful because they don’t have to use as much of their energy reserves being vigilant?" I think you only addressed the second portion of their original comment and changed the next sentence to say additionally instead of in this way, but you didn't finish your thought.

Due to the corrected analysis, we removed text regarding this. 

Lines 333-335: see previous comment about your interspecific density metric

The interspecific density metric has been reworked and clarified. 

Lines 344-349: I see this was added to address Reviewer 2's comments. I think this does that, but could be re-worded for clarity using what the hypothesis is rather than referring to rejecting your hypothesis and having the reader hunt back through the text for what the hypothesis was exactly.

We added text to clarify what hypothesis was rejected and why. 

Lines 350-353: I think I understand what you mean, but consider re-wording for clarity, as to what the considerations should be- just that managers know there are birds that are non-breeders or migrants? And not assume the population is doing as well? What's the implication?

This has been reworded for clarity. 

Lines 358-359: consider re-wording "The exact mechanisms that caused the differences for Brewer's sparrows" reads odd. Which differences? In nest density?

This has been reworded for clarity. 

Line 360-362: perhaps you are missing a word here? I think you are trying to say that Nest density and nest success did not differ for the thick billed longspur

This has been reworded for clarity. 

Line 362: similarly

This has been reworded for clarity. 

Lines 360-363: Maybe a sentence or a few words connecting these thoughts that other species in the area were not affected by SGI status.

We added text to connect these thoughts and reference similar studies in the area regarding greater sage-grouse. 

Lines 395-397: I'm not sure you need this statement. Now that the paper is less focused on grazing it seems a bit out of context.

This line was removed as suggested by the reviewer.

---

## [Editor Report · Decision Letter 2]

24 Jul 2023

Density dependence of songbird demographics in grazed sagebrush steppe

PONE-D-22-21628R2

Dear Dr. Ruth,

We’re pleased to inform you that your manuscript has been judged scientifically suitable for publication and will be formally accepted for publication once it meets all outstanding technical requirements.

Kind regards,

Karen Root, Ph.D.

Academic Editor

PLOS ONE

Additional Editor Comments (optional):

I appreciate the diligence and persistence of the authors in multiple revisions of this manuscript. The authors have been thorough and addressed all of the concerns of the previous peer reviews. The tables and figure in the online appendices.docx are particularly helpful and improve clarity and strengthen the conclusions, although the comments should be removed. This paper increases our understanding of density dependence and vital rates of two grassland bird species in a highly impacted landscape.
---

## [Editor Report · Acceptance letter]

7 Aug 2023

PONE-D-22-21628R2 

Density dependence of songbird demographics in grazed sagebrush steppe 

Dear Dr. Ruth:

I'm pleased to inform you that your manuscript has been deemed suitable for publication in PLOS ONE. Congratulations! Your manuscript is now with our production department. 

Kind regards, 

on behalf of

Professor Karen Root 

Academic Editor

PLOS ONE